# A Survey on Efficient Protein Language Models

**Shouren Wang**[1] **Debargha Ganguly**[1] **Vinooth Rao Kulkarni**[1] **Wang Yang**[1]
**Zhuoran Qiao**[2] **Daniel Blankenberg**[3] **Vipin Chaudhary**[1] **Xiaotian Han**[1]

[1] *Case Western Reserve University*  [2] *Chai Discovery*  [3] *Cleveland Clinic*

*{sxw992, dxg512, vxk285, wxy320, vipin, xhan}@case.edu, zrqiao0@gmail.com, blanked2@ccf.org*

**Reviewed on OpenReview:** *https://openreview.net/forum?id=PTReuOwsXz*

## Abstract

Protein language models (pLMs) have become indispensable tools in computational biology, driving advances in variant effect prediction, functional annotation, structure prediction, and engineering. However, their rapid expansion from millions to tens of billions of parameters introduces significant computational, accessibility, and sustainability challenges that limit practical application in environments constrained by GPU memory, hardware availability, and energy budgets. This survey presents the first comprehensive review of efficient pLMs, synthesizing recent advancements across four key dimensions. We first examine *(1)* dataset efficiency through meta-learning-based few-shot and scaling-law-guided data allocation; and *(2)* architecture efficiency via lightweight alternatives including quantized transformers, embedding compression, and convolution-based designs. Furthermore, we review *(3)* training efficiency through scaling-law-informed pretraining, structure-integrated multimodal approaches, and low-rank adaptations with diverse distillation strategies; and *(4)* inference efficiency via quantization, dense-retrieval, and structure-search methods. By providing a structured taxonomy and practical guidance, this survey enables the development of high-performance, scalable, yet sustainable next-generation pLMs. We additionally trace the historical evolution of pLMs from early sequence models to modern multi-billion-parameter architectures, analyze intersections across efficiency dimensions, identify evaluation comparability challenges, discuss LLM techniques not yet explored for pLMs, and offer practical recommendations for selecting efficiency strategies. A companion repository for this survey—containing the figures, a categorized reading list, and regularly updated releases—is available at https://github.com/SR-A-W/efficient-protein-language-model-survey.

## 1 Introduction

Inspired by the success of large language models (LLMs) in natural language processing (NLP) (Vaswani et al., 2017; Devlin et al., 2019), protein language models (pLMs) have emerged as a transformative force in computational biology (Bepler & Berger, 2019). By treating AA sequences as a "language of life," (Asgari & Mofrad, 2015) pLMs apply self-supervised learning to vast protein databases to learn the fundamental grammar of protein structure, function, and evolution (Rives et al., 2021; Elnaggar et al., 2021). A comparative overview between LLMs and pLMs is shown in Figure 3, providing intuitive context for their methodological parallels and distinctions. This paradigm shift enables protein analysis at an unprecedented scale of hundreds of millions of sequences (Lin et al., 2023). Rather than serving as a direct replacement for experimental methods such as X-ray crystallography, NMR spectroscopy, and Cryo-EM, pLMs complement these techniques by enabling tasks that are infeasible at experimental throughput: predicting the functional effects of mutations across entire protein families (Meier et al., 2021), annotating protein function directly from sequence (Rives et al., 2021), and predicting three-dimensional folds for hundreds of millions of sequences in days rather than decades (Lin et al., 2023). The power of pLMs lies in their ability to bridge the enormous gap between the over 200 million known protein sequences and about 250 thousand experimentally determined structures (Burley et al., 2021; Consortium, 2021), thereby enabling concrete advances in drug discovery, such

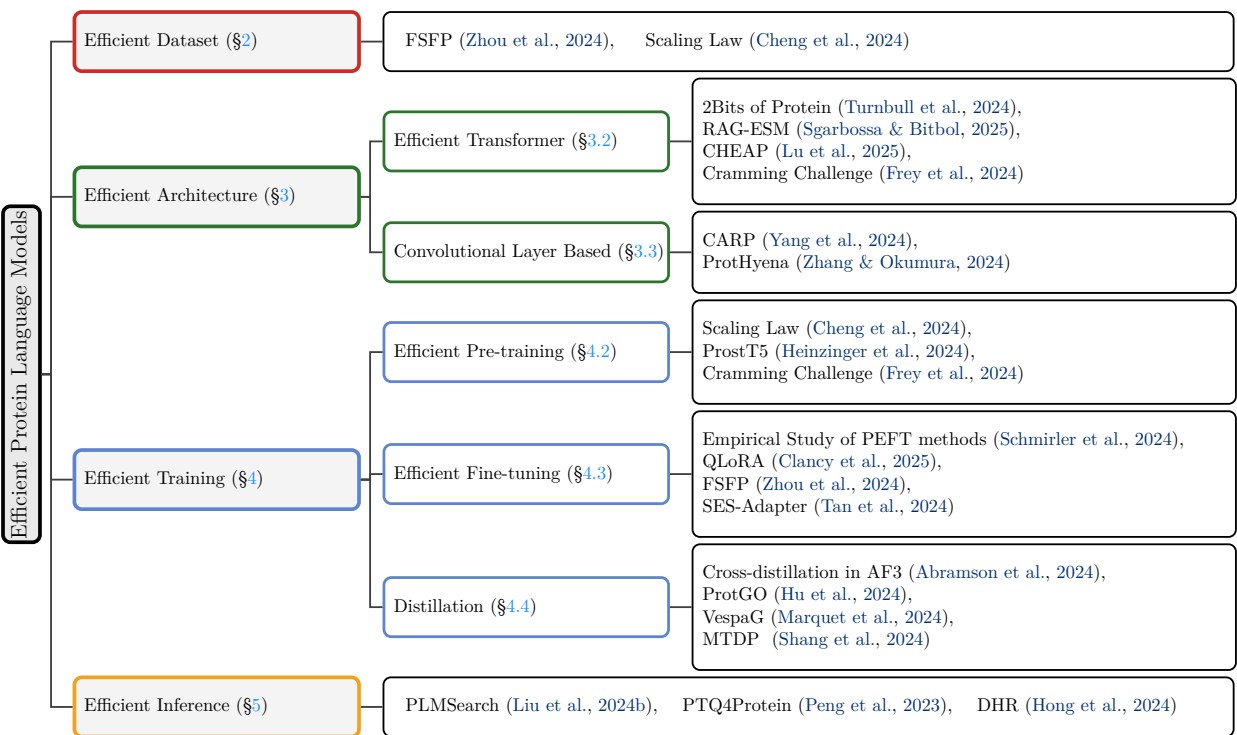

Figure 1: An overview and taxonomy of the efficient Protein Language Model methodologies discussed in this survey. We structure our review into four primary pillars—Dataset, Architecture, Training, and Inference—and highlight the key approaches and representative works within each category.

as virtual screening of candidate binders, computational guidance of directed evolution campaigns (Buller et al., 2018), and functional annotation of previously uncharacterized proteins (Xiao et al., 2025), as well as synthetic biology. A historical overview of key milestones in pLM development is summarized in Figure 2.

The remarkable capabilities of pLMs stem from their deep neural architectures, predominantly transformer-based networks (Vaswani et al., 2017), which are trained extensively on massive protein sequence datasets (Suzek et al., 2015) to learn intrinsic biological patterns and representations. The entire pipeline of pLMs consists of dataset construction, model architecture design, training methodologies (pre-training, fine-tuning, and distillation), and inference strategies, each influencing both performance and efficiency (Ofer et al., 2021; Ferruz et al., 2022). Powered by large-scale computational resources, these models integrate evolutionary and structural information from massive-scale evolutionary protein corpora into parameter spaces reaching up to tens of billions (Lin et al., 2023), thereby achieving state-of-the-art accuracy on structure-prediction benchmarks such as CAMEO (Haas et al., 2013) and CASP14 (Kryshtafovych et al., 2021), a level not previously reached by single-sequence models.

Despite their transformative potential, the effectiveness of large-scale pLMs comes at a substantial computational and memory cost, posing significant practical challenges. Typical large pLMs contain billions of parameters, such as the 15B parameter ESM-2 (Lin et al., 2023), and require extensive resources; for instance, training models of this scale conventionally demands tens of thousands of GPU hours, translating to immense financial and environmental burdens (Lin et al., 2023; Strubell et al., 2019; 2020). This heavy reliance on compute limits accessibility for resource-constrained research groups (Kaplan et al., 2020). Furthermore, deploying such extensive models in real-world scenarios, such as proteome-wide homology searches, becomes computationally formidable, restricting their broader applicability and utility (Hong et al., 2024).

Importantly, the benefits of scaling pLMs are not always clear-cut. Many production use cases, including variant effect prediction and protein fitness landscape estimation, achieve respectable performance with sub-billion parameter models (e.g., ESM-2 at 8M, 35M, or 150M parameters), without any particular care for extreme efficiency (Lin et al., 2023; Meier et al., 2021). Scaling benefits often plateau for certain downstream

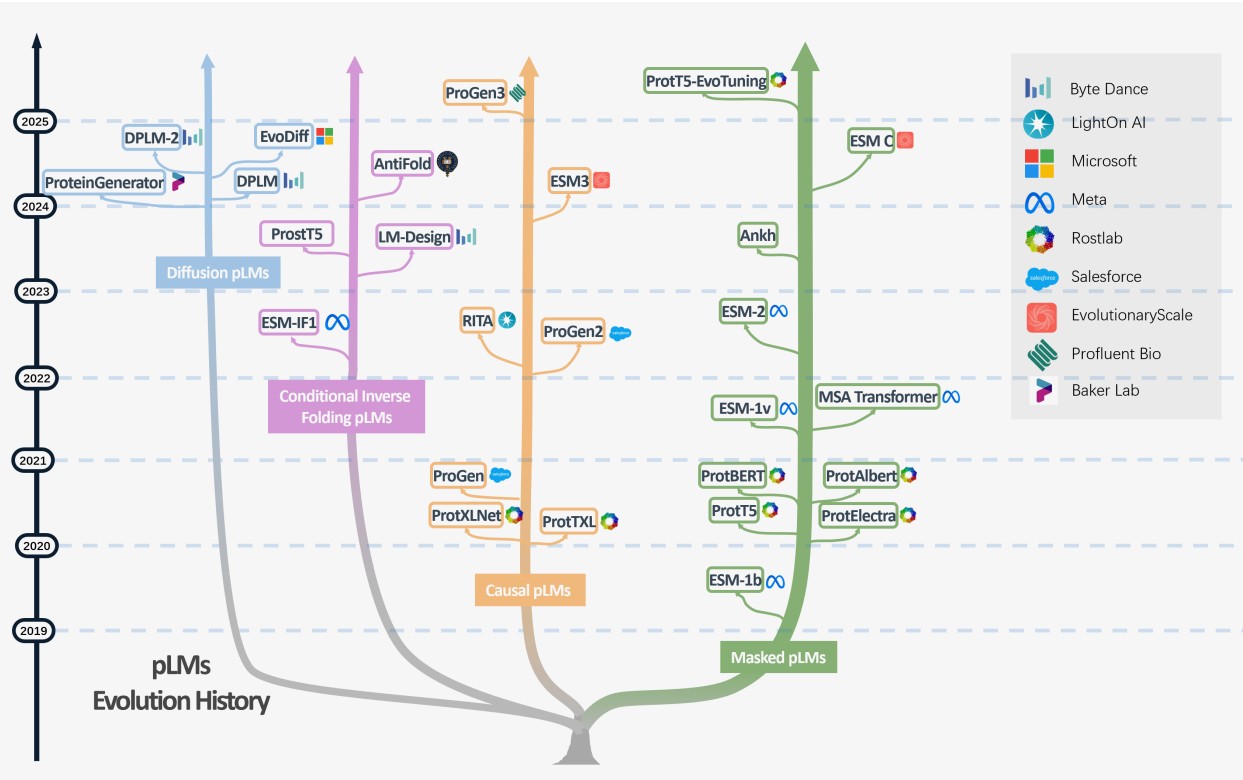

Figure 2: The evolution history of Protein Language Models (pLMs) from 2019 to 2025. The timeline illustrates the development and divergence of major pLM families, such as Masked pLMs and Conditional Inverse Folding pLMs, highlighting key models like the ESM series, ProtT5, and ProGen, along with their originating research labs.

tasks, suggesting that larger is not always better (Frey et al., 2024). Moreover, specialized scientific foundation models such as pLMs are increasingly being leveraged as components within more complex agentic workflows, for example, autonomous protein design loops that iteratively query a pLM for fitness predictions, propose mutations, and validate candidates (Xiao et al., 2025). Such iterative pipelines amplify the importance of efficient models capable of fast, low-latency inference. These observations further motivate the efficiency-focused perspective adopted in this survey.

Motivated by these challenges, a growing body of research has focused explicitly on improving the efficiency of pLMs across various stages of their lifecycle (Cheng et al., 2024; Zhou et al., 2024; Frey et al., 2024). These *efficiency-focused* methods aim to achieve competitive or even superior performance while reducing resource consumption by orders of magnitude. Specifically, efficiency strategies in pLMs encompass several critical categories, the reviewed methods of which are listed in Figure 1:

- **Dataset Efficiency**: Enhancing pLM efficiency via optimizing the allocation and utilization of datasets (Cheng et al., 2024), to maximize learning from limited or expensive experimental data (Zhou et al., 2024).

- **Architecture Efficiency**: Developing compact or lightweight architectures, such as quantized transformer models (Turnbull et al., 2024) or convolutional architectures (Yang et al., 2024; Zhang & Okumura, 2024), reducing parameter counts by up to 10× while maintaining predictive performance.

- **Training Efficiency**: Optimizing the training procedures, including computational resource allocation guided by scaling laws (Cheng et al., 2024), efficient fine-tuning techniques (Schmirler et al., 2024; Clancy et al., 2025; Zhou et al., 2024), and knowledge distillation (Hinton et al., 2015) approaches that transfer knowledge from larger, teacher models to compact, efficient student models (Abramson et al., 2024; Hu et al., 2024; Marquet et al., 2024; Shang et al., 2024).

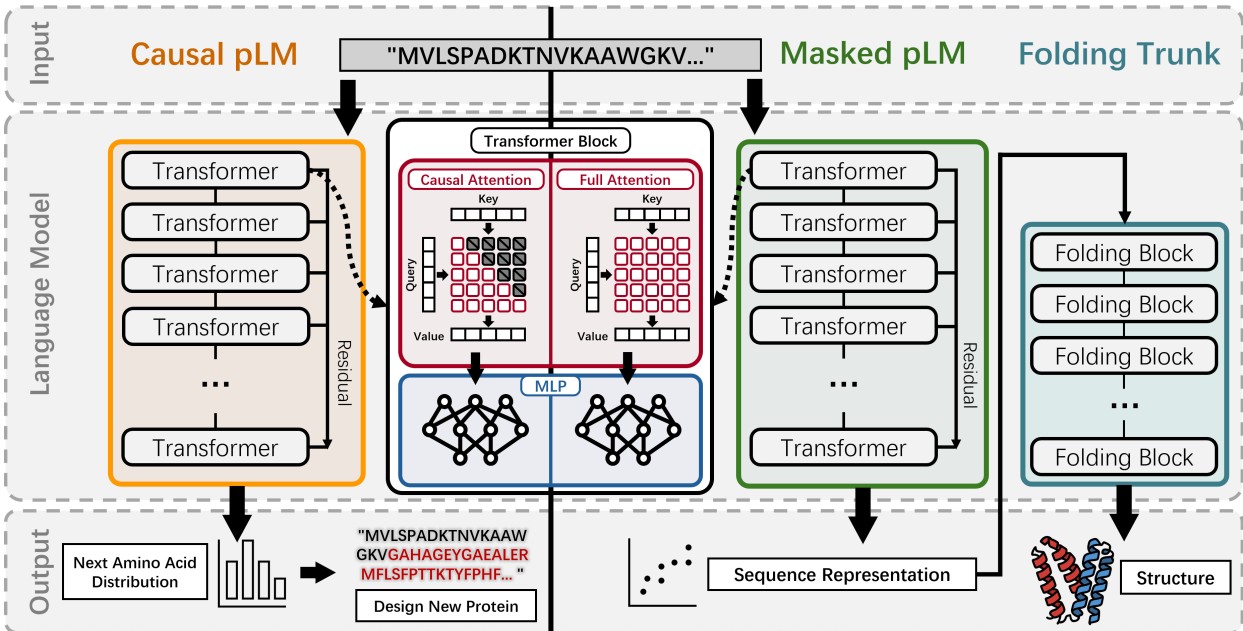

Figure 3: A comparative overview of **Causal Protein Language Models (Causal pLMs)** and **Masked Protein Language Models (Masked pLMs)**. While both architectures are rooted in the Transformer framework, they serve distinct biological objectives. Causal pLMs (left) utilize causal attention for autoregressive sequence generation, modeling the probability of the next amino acid for *de novo* design. In contrast, Masked pLMs (right) employ full attention to learn contextual sequence representations from amino acid inputs, which facilitate downstream tasks such as structure prediction via dedicated folding blocks (e.g., as in ESMFold (Lin et al., 2023)).

- **Inference Efficiency**: Employing techniques such as post-training quantization (PTQ) (Peng et al., 2023), dense embedding-based retrieval (Liu et al., 2024b), and alignment-free methods to accelerate search by over two orders of magnitude compared to traditional alignment methods (Hong et al., 2024).

While recent surveys have provided broad overviews of protein language models (pLMs) and their applications (Xiao et al., 2025; Wu et al., 2022), our work particularly focuses on the efficiency dimension of pLMs across dataset, architecture, training, and inference. We systematically review these efficiency-focused methodologies, highlighting their core principles, underlying mechanisms, and performance implications. The remainder of this survey is organized as follows: Section 1.1 provides a historical overview of pLM development and key applications, Section 2 details dataset strategies, Section 3 reviews architectural innovations, Section 4 discusses training methodologies, Section 5 covers efficient inference methods, Section 6 presents a cross-cutting discussion of evaluation, intersections across efficiency dimensions and Section 7 outlines underexplored and future directions.

## 1.1 Evolution and Applications of Protein Language Models

This sub-section provides a quick historical overview of pLM development, focusing on scaling of data, models and tasks, while summarizing the key application areas that have emerged from this rapidly evolving field, complementing the timeline depicted in Figure 2.

### 1.1.1 Scaling & Evolution of pLMs (2019–2025)

The development of protein language models can be organized into several overlapping phases, each marked by architectural innovations, scaling advances, and broadening scope. The concept of learning transferable protein representations from unlabeled sequences crystallized in 2019, when multiple groups independently

demonstrated that self-supervised language models trained on large protein databases could produce embeddings useful for downstream tasks such as stability prediction, function annotation, and contact prediction.

**The masked-language-model era (2020–2021).** Inspired by BERT (Devlin et al., 2019), the masked language modeling (MLM) objective became the dominant pre-training strategy for pLMs. ESM-1b (Rives et al., 2021), a 650M-parameter transformer trained on UniRef50, showed that attention maps in deep protein transformers implicitly encode residue–residue contacts, linking language-model internals directly to three-dimensional structure. ProtTrans (Elnaggar et al., 2021) systematically benchmarked six architectures, including BERT, ALBERT, XLNet, T5, and Electra variants (ProtBERT, ProtAlbert, ProtXLNet, ProtT5, ProtElectra, ProtTXL), trained on UniRef and BFD at scales up to 11 billion parameters, with ProtT5-XL-U50 emerging as one of the most widely adopted pLMs for generating per-residue embeddings. The MSA Transformer (Rao et al., 2021) extended the masked language modeling paradigm from single sequences to multiple sequence alignments, showing that a model operating over aligned homologous sequences captures coevolutionary patterns more effectively and yields state-of-the-art unsupervised contact predictions. ESM-1v (Meier et al., 2021) demonstrated that a single masked language model, without any task-specific fine-tuning, could predict the functional effects of missense mutations in a zero-shot manner, achieving performance competitive with supervised and evolution-based methods across deep mutational scanning datasets.

**Scaling, structure, and diversification (2022–2023).** This period witnessed three major trends: aggressive scaling of masked pLMs, the integration of structural information, and the rise of autoregressive and diffusion-based generative models.

On the scaling front, ESM-2 (Lin et al., 2023) systematically trained a family of transformer models from 8M to 15B parameters and demonstrated smooth scaling of structure prediction quality with model size. Ankh (Elnaggar et al., 2023) further explored the efficiency–performance frontier by optimizing ProtT5's training protocol and demonstrating that careful hyperparameter tuning can yield competitive models at reduced computational budgets. Structure-aware pLMs emerged as a distinct subfield. ProstT5 (Heinzinger et al., 2024) adopted a bilingual strategy, training a T5-based model to translate between amino acid sequences and 3Di structural sequences, thereby embedding structural knowledge without requiring explicit 3D coordinates at inference time.

On the generative side, autoregressive models demonstrated increasing capabilities. ProGen (Madani et al., 2023) showed that autoregressive transformers conditioned on functional tags could generate artificial proteins with measurable enzymatic activity, directly validating the capacity of pLMs for controllable protein design. ProGen2 (Nijkamp et al., 2023) scaled this approach to 6.4 billion parameters across multiple training datasets, showing improved fitness prediction and generation quality with scale. RITA (Hesslow et al., 2022) explored autoregressive protein language models up to 1.2 billion parameters, demonstrating competitive performance on variant effect prediction and providing insights into the scaling behavior of generative pLMs.

**Diffusion and structure-conditioned generation (2023–2024).** A complementary line of work explored diffusion-based and structure-conditioned approaches to protein generation. EvoDiff (Alamdari et al., 2023) introduced discrete diffusion models for protein sequences, enabling unconditional and evolution-guided generation directly in sequence space without requiring structural information. ProteinGenerator (Lisanza et al., 2025) combined sequence and structure generation using a joint diffusion framework, producing *de novo* proteins with high experimental success rates.

For the inverse folding problem, designing amino acid sequences that fold into a given three-dimensional backbone structure, ESM-IF1 (Hsu et al., 2022) demonstrated that a GVP-Transformer (Hsu et al., 2022) trained with an inverse-folding objective could produce useful fitness predictions through fixed-backbone scoring, connecting structure-conditioned generation with variant effect prediction. LM-Design (Zheng et al., 2023) leveraged pre-trained pLMs for structure-based sequence design by iteratively refining sequences using the knowledge encoded in masked language models, achieving competitive results with dedicated inverse-folding architectures. AntiFold (Høie et al., 2025) specialized the inverse folding paradigm for antibody design, training on antibody-specific structures to generate diverse, designable complementarity-determining region sequences.

**Recent developments (2024–2025).** The most recent phase is characterized by unification, multimodality, and continued scaling. ESM3 (Hayes et al., 2025) scaled flops 25x from its prior version and introduced a multimodal generative model scaling up to 98B parameters, that reasons jointly over protein sequence, structure, and function using a masked generative framework, demonstrating the ability to generate functional proteins, including a novel green fluorescent protein, through iterative decoding across modalities. DPLM (Wang et al., 2024) formulated protein language modeling as discrete diffusion, combining the generative flexibility of diffusion processes with the rich representations of pre-trained pLMs. DPLM-2 (Wang et al., 2025a) extends this discrete diffusion framework by introducing Lookup-Free Quantization (LFQ) to better incorporate three-dimensional structural information, improving structure consistency and inverse-folding quality. Complementing this model-centric line, Hsieh et al. (2025) systematically study the broader design space of multimodal protein language models, identifying tokenization loss and inaccurate structure-token prediction as key bottlenecks and showing that advances in generative modeling, structure-aware architectures and representation learning, and data exploration can substantially strengthen structural modeling.

ProGen3 (Bhatnagar et al., 2025) redefines the scale of protein modeling by deploying a sparse Mixture-of-Experts (MoE) architecture; this allows the 46B-parameter model to activate only 27% of its capacity per inference, achieving a 3x throughput increase (w.r.t dense models) while training on a massive 1.1-1.5 trillion amino acid tokens. On the other hand, ESM Cambrian (Team, 2024) represents a new generation of protein language models that achieve state-of-the-art biological representations by reducing scale to 6 billion parameters—outperforming the legacy 15-billion-parameter ESM-2 model. By unifying sequence, structure, and functional annotations into a single pre-training framework, ESM Cambrian delivers a 3x to 4x increase in throughput and enables a 600M-parameter model to rival the accuracy of previous 3B-parameter architectures. Concurrently, the community has shifted toward such "efficient" pLMs that maximize biological depth per FLOP, motivating the core focus of this survey.

### 1.1.2 Scaling of Key Applications of pLMs

The representations and generative capabilities learned by pLMs have enabled a broad spectrum of downstream applications, which we organize into four categories. Notably, each application area has itself grown in complexity alongside the models, progressing from simple downstream classification toward increasingly ambitious generative and structure-aware tasks.

- **Structure prediction.** Perhaps the most visible success of pLMs is in protein structure prediction. Early masked models such as ESM-1b (Rives et al., 2021) revealed that transformer attention maps implicitly encode residue–residue contacts, providing coarse structural signals from sequence alone. This capability matured with ESMFold (Lin et al., 2023), which leverages ESM-2 embeddings to predict full three-dimensional structures from single sequences without requiring multiple sequence alignments, enabling proteome-scale structure prediction at orders-of-magnitude faster speeds than alignment-dependent methods such as AlphaFold2 (Jumper et al., 2021). These models have been applied to annotate the structural landscape of entire metagenomic databases, revealing novel protein folds absent from the Protein Data Bank.

- **Variant effect and fitness prediction.** Predicting the functional consequences of amino acid substitutions is critical for clinical genetics and protein engineering. Early approaches relied on supervised models trained on labeled mutagenesis data, but zero-shot scoring with masked pLMs shifted the paradigm: ESM-1v (Meier et al., 2021) and ESM-2 (Lin et al., 2023) achieve competitive performance on deep mutational scanning benchmarks without any supervised training data. More recently, structure-aware models such as SAProt (Su et al., 2024) and inverse-folding models such as ESM-IF (Hsu et al., 2022) have further improved prediction accuracy by incorporating three-dimensional context, reflecting the broader trend toward integrating sequence and structure. The ProteinGym benchmark (Notin et al., 2022) has become the standard evaluation platform for this task, aggregating hundreds of deep mutational scanning assays.

- **Protein generation and design.** Protein generation has progressed from unconditional sequence sampling to increasingly controlled and structure-aware design. Early autoregressive pLMs such as ProtGPT2 (Ferruz et al., 2022) sample from learned sequence distributions, while ProGen (Madani et al., 2023) and ProGen2 (Nijkamp et al., 2023) introduced functional conditioning through control tags, enabling the

Table 1: Summary of representative dataset-related methods for pLMs.

| Category | Method | Brief Description | textbfLimitation |
|---|---|---|---|
| Dataset | Scaling-law (Cheng et al., 2024) | Empirical study on optimal data/model allocation for pLM training. | Become "unruly" in data-constrained environments, requiring strategies like model reusing to restore predictable performance gains and improve data efficiency. |
| | FSFP (Zhou et al., 2024) | few-shot with meta-learning and LoRA (Hu et al., 2022) under severe data scarcity. | Heavy dependence on the availability of similar evolutionary data for auxiliary tasks & LoRA's inherent low-rank capacity bottleneck, which often fails to capture complex biological landscapes under severe data scarcity. |

generation of artificial enzymes with experimentally validated activity. On the structure-conditioned side, ProteinMPNN (Dauparas et al., 2022) designs sequences for given backbone geometries with high experimental success rates, and has been widely adopted as a core module in computational design pipelines. Most recently, multimodal models such as ESM3 (Hayes et al., 2025) unify sequence, structure, and function in a single generative framework, representing a shift toward joint reasoning across modalities.

- **Representation learning and transfer.** A foundational use case for pLMs is to extract per-residue or per-sequence embeddings that serve as feature inputs for downstream classifiers and regressors. Early single-modality models such as ESM-1b (Rives et al., 2021) and ProtT5 (Elnaggar et al., 2021) encode secondary structure, subcellular localization, binding site identity, and protein function with high accuracy upon simple linear probing or lightweight fine-tuning. ProteinBERT (Brandes et al., 2022) demonstrated that joint pre-training on sequence and functional annotations improves transfer performance on Gene Ontology prediction tasks. More recent models such as Ankh (Elnaggar et al., 2023) and ESM Cambrian (Team, 2024) push this further by delivering stronger representations at reduced computational cost, reflecting the field's growing emphasis on efficiency.

Collectively, these applications underscore the versatility of pLMs as general-purpose tools for computational protein science. At the same time, the growing complexity and scale of both models and applications amplify the need for efficient methods across the pLM lifecycle, the central theme of the subsequent sections.

## 2 Datasets for Efficient pLMs

### 2.1 Background and Challenges

The training paradigm for modern language models is fundamentally data-driven, predicated on learning statistical and semantic regularities from vast corpora (Kaplan et al., 2020). For Large Language Models (LLMs) like BERT (Devlin et al., 2019) and the GPT series (Radford et al., 2018; Brown et al., 2020), this involves processing immense textual datasets that are rigorously deduplicated, filtered, and ultimately discretized into a finite vocabulary via tokenization (Sennrich et al., 2016; Kudo & Richardson, 2018). Transposing this paradigm to computational biology, Protein Language Models (pLMs) operate on datasets of AA sequences drawn from large-scale biological repositories such as UniRef (Suzek et al., 2015) and expansive metagenomic collections (Mitchell et al., 2020). These biological datasets undergo analogous preprocessing, including stringent sequence identity filtering and task-specific curation, to construct a suitable training corpus (Cheng et al., 2024).

Within both NLP and computational biology, an efficiency-centric approach to dataset construction seeks to maximize model utility and generalization performance under a constrained computational-budget. For LLMs, this often involves curating data for optimal scale and diversity to avoid diminishing returns on compute (Hoffmann et al., 2022; Raffel et al., 2020). In the context of pLMs, efficiency-driven data strategies have emerged to address distinct biological challenges. Key approaches include meta-learning frameworks for few-shot adaptation, which are critical in scenarios where experimental labels are prohibitively expensive or scarce (Zhou et al., 2024). Additionally, researchers have formulated empirically-derived scaling laws that govern the co-scaling of model and data size to optimize the training trajectory within a fixed computational

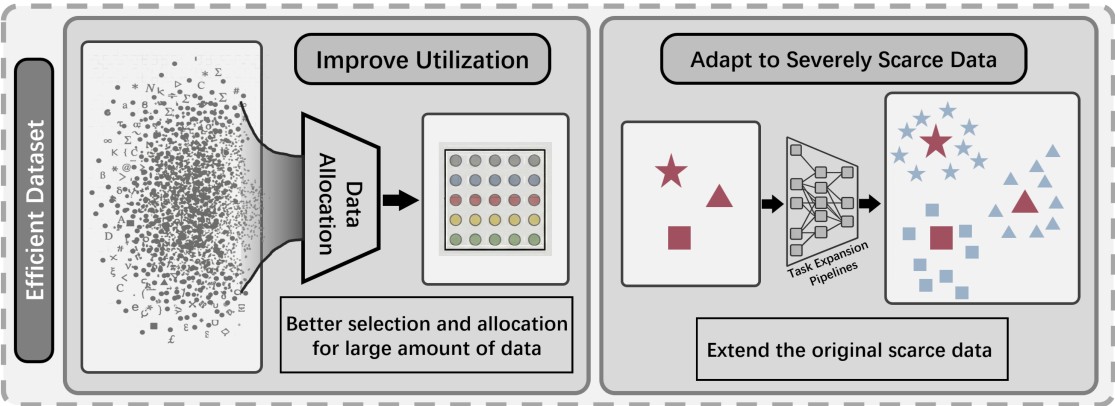

Figure 4: Conceptual illustration of efficient dataset strategies. The left panel, 'Improve Utilization,' depicts the systematic selection and allocation of data from large-scale corpora guided by scaling laws to optimize training efficiency. The right panel, 'Adapt to Severely Scarce Data,' illustrates the extension of limited experimental datasets through task expansion pipelines and auxiliary task construction, enabling robust few-shot adaptation in data-constrained scenarios.

envelope (Cheng et al., 2024). Such methods directly confront data-related bottlenecks, aiming to enhance model performance within the practical constraints of bioinformatics research.

The development of these efficiency-focused strategies is motivated by several fundamental challenges inherent to biological data utilization:

- **Paucity of Labeled Data:** A critical bottleneck in many real-world applications, such as protein engineering and functional genomics, is the extreme scarcity of experimentally validated labels. Often, only a few dozen labeled protein variants are available, posing a fundamental impediment to traditional supervised fine-tuning and demanding highly data-efficient adaptation methods (Zhou et al., 2024).

- **Absence of Principled Scaling Guidelines:** Given a finite computational-budget, determining the optimal allocation between model parameter count and training data volume remains a significant challenge. An unprincipled approach, such as aggressively scaling model size while repeatedly cycling through a small dataset, can lead to suboptimal training dynamics, including premature convergence, overfitting, and a failure to capitalize on the model's architectural capacity (Cheng et al., 2024).

A closely related challenge is the pronounced *sequence–structure imbalance*: while over 200 million protein sequences are catalogued in UniProt (Consortium, 2021), about 250,000 have experimentally determined structures (Burley et al., 2021). This mismatch is particularly relevant for training multimodal pLMs that seek to jointly leverage sequence and structural information. Several strategies have emerged to address this gap. ProstT5 (Heinzinger et al., 2024) sidesteps the scarcity of experimental structures by using AlphaFoldDB-predicted structures (Varadi et al., 2024), encoded as 3Di token sequences (Van Kempen et al., 2024), to provide structural supervision at the scale of hundreds of millions of proteins. SES-Adapter (Tan et al., 2024) injects structural knowledge into frozen pLM backbones via cross-modal attention over structure representations serialized from predicted structures, thereby augmenting sequence-only models without requiring additional experimental data. FSFP (Zhou et al., 2024) takes an alternative route, using MSA-derived evolutionary signals (Laine et al., 2019) as a proxy for structural and functional information under extreme data scarcity. Together, these approaches demonstrate that the sequence–structure gap can be effectively bridged through computational structure prediction, cross-modal adapters, and evolutionary proxy signals, enabling multimodal pLM training even in the absence of large-scale experimental structure data.

To surmount these obstacles, recent work has focused on developing sophisticated meta-learning protocols and establishing a principled, theoretical foundation for balancing model and data scaling. The subsequent section reviews these methodologies in detail, with an overview provided in Table 1. A conceptual illustration of these efficient dataset strategies is provided in Figure 4.

## 2.2 Efficient Dataset Methods

**Few-Shot Learning for Protein Fitness Prediction (FSFP)** (Zhou et al., 2024) addresses the challenge of efficiently fine-tuning pLMs under severe data scarcity (Romero et al., 2013), enabling fitness-prediction with only tens of experimentally labeled mutants per target protein. The key innovation of FSFP lies not in increasing data volume, but in its auxiliary-task construction, evolutionary-informed pseudo-labeling, and meta-learning-driven adaptation, providing a robust solution for pLM fine-tuning in low-data scenarios.

FSFP frames fitness-prediction as a ranking task, employing a listwise-loss (Cao et al., 2007) throughout both meta-learning and fine-tuning, which better reflects practical needs in protein engineering research and applications (Buller et al., 2018). FSFP consists of the following major components:

- *Auxiliary Task Construction:* FSFP constructs three auxiliary tasks: it first retrieves two labeled DMS datasets (Fowler & Fields, 2014; Melamed et al., 2013) from proteins similar to the target, and additionally generates a third pseudo-labeled dataset using GEMME (Laine et al., 2019), which leverages evolutionary information from MSA (Thompson et al., 1994; Mirdita et al., 2022).

- *Meta-Learning:* These auxiliary tasks are used for meta-training via MAML (Finn et al., 2017), equipping the pLM for rapid adaptation.

- *Parameter-Efficient Adaptation:* To prevent overfitting under few-shot settings (Snell et al., 2017), FSFP freezes the backbone pLM and trains LoRA (Hu et al., 2022) parameters; meta-learning yields a better LoRA initialization that the subsequent target-task adaptation continues to update.

Benchmarking on 87 protein datasets (Notin et al., 2023), FSFP demonstrates superior performance compared to strong baselines. It outperforms unsupervised zero-shot inference (including ESM-1v (Meier et al., 2021), ESM-2(Lin et al., 2023), and Saprot (Su et al., 2024)) and significantly surpasses the state-of-the-art supervised few-shot baseline, Ridge Regression (Hsu et al., 2022). FSFP boosts the average spearman correlation by up to 0.1 and raises positive hit rates in real-world engineering by 25% (Zhou et al., 2024).

**Scaling laws for dataset allocation in pLM training** (Cheng et al., 2024) have been empirically established to maximize training efficiency under fixed compute. Similar to LLMs in NLP tasks, simply increasing model size or compute is insufficient (Kaplan et al., 2020; Hoffmann et al., 2022); the size *and* diversity of training data are critical. Empirical analysis shows that masked language models (MLMs) (Devlin et al., 2019) and causal language models (CLMs) (Radford et al., 2019; Brown et al., 2020) follow distinct power-law relationships between model size and required data, with sublinear data scaling (Hestness et al., 2017). Repeating small datasets across many epochs yields overfitting in MLMs (Roelofs et al., 2019) and diminishing returns in CLMs.

To address these issues, the study introduces the *UniMeta200B* dataset (Mitchell et al., 2020; Nayfach et al., 2021), aggregating diverse metagenomic protein sequences with strict deduplication. This improves out-of-domain perplexity and yields more stable independent and identically distributed validation curves with reduced variance across families.

The resulting scaling laws provide explicit formulas for allocating parameters and dataset size under a compute constraint, enabling principled, data-centric decisions in pLMs development. For fitted relations and resource-allocation details, see Section 4.2. These results highlight that enhancing dataset diversity, rather than simply expanding model size or repeating data, is essential for efficient protein language modeling.

> **Takeaway in §2 for Efficiency-focused pLMs Dataset Methods**
>
> Dataset efficiency-focused strategies—meta-learning-based few-shot and principled scaling laws—address two key bottlenecks for pLMs: label scarcity and compute constraints. They are especially critical because protein labels require costly experimental assays, making data scarcity a hard physical constraint rather than an annotation bottleneck. Crucially, the evolutionary relatedness of protein families provides a compensating prior with no NLP analogue: conserved fitness landscapes enable few-shot transfer across homologs, while family-level redundancy and the sequence–structure imbalance drive masked and causal LMs (MLMs and CLMs) to follow distinct scaling exponents that differ both from each other and from their NLP counterparts, demanding domain-specific data allocation.

Table 2: Summary of representative efficient model architectures for pLMs.

| Category | Method | Brief Description | Limitation |
|---|---|---|---|
| Transformer-based | 2Bits of Protein (Turnbull et al., 2024) | Training with ternary weights for memory and compute efficiency. | Performance gap in smaller models that only closes at a 3-billion-parameter scale, requiring higher-precision embedding and output layers to maintain stability. |
| | RAG-ESM (Sgarbossa & Bitbol, 2025) | Lightweight retrieval-augmented model with parameter-sharing. | Performance significantly decreases as the evolutionary distance between the input and retrieved context sequences increases, high sensitivity to retrieval quality |
| | CHEAP (Lu et al., 2025) | Compressing high-dimensional embeddings while preserving structure. | Maintaining structural fidelity at high compression ratios, with structure proving significantly harder to compress than sequence information. |
| | Cramming Challenge (Frey et al., 2024) | Train a performant pLM in 24 hours on a single GPU. | Architectural and pooling choices must be fixed globally for all downstream tasks, restricting task-specific optimization. |
| CNN-based | CARP (Yang et al., 2024) | Using dilated-convolutions (ByteNet-style) to replace attention layers. | Fixed receptive field that requires specific tuning for sequence length, unlike Transformers. |
| | Hyena operator (Zhang & Okumura, 2024) | Implicit long convolutions for sub-quadratic scaling on long sequences. | Long convolutions lack natural permutation equivariance and that distilled recurrent forms may lose qualitative information |

# 3 Efficient Architecture

## 3.1 Background and Challenges

Model architecture fundamentally determines the representational capability and computational complexity of language models. LLMs in NLP predominantly employ transformer architectures (Vaswani et al., 2017), characterized by self-attention mechanisms that effectively capture long-range-dependencies. However, the quadratic scaling of transformers in computational complexity and memory consumption with respect to input sequence length poses significant efficiency challenges (Vaswani et al., 2017; Hoffmann et al., 2022). Alternatively, convolutional neural networks (CNNs) (Kalchbrenner et al., 2016; Gehring et al., 2017), leveraging local receptive fields, scale linearly with sequence length and inherently incorporate positional information through sliding window operations. These distinct architectural paradigms have informed similar developments in pLMs, where transformers remain dominant due to their robust performance on capturing long-range residue interactions (Rives et al., 2021), while CNN-based methods have emerged as efficient alternative (Yang et al., 2024).

Efficiency-focused model architectures in LLMs typically include strategies such as weight quantization, embedding dimension reduction, and structural sparsity to mitigate computational overheads (Tay et al., 2022; Frantar et al., 2022; Frankle & Carbin, 2018). Analogously, recent advancements in pLM architectures have pursued quantization of weights (Turnbull et al., 2024), embedding compression (Lu et al., 2025), and cross-attention (Vaswani et al., 2017) modules conditioned on homologous sequences (Sgarbossa & Bitbol, 2025) to integrate evolutionary information efficiently (Hopf et al., 2017; Jumper et al., 2021). Convolution-based models, with inherent linear scaling, further enhance computational efficiency, presenting competitive alternatives for large-scale protein modeling tasks (Yang et al., 2024).

The emergence of efficiency-focused model architectures in pLMs is largely driven by several challenges in model architecture, including:

- **High Computational and Memory Cost:** Standard transformer-based pLMs require quadratic time and memory with respect to sequence length, leading to prohibitive resource demands for long proteins or large models. The use of full-precision weights and high-dimensional embeddings further increases GPU

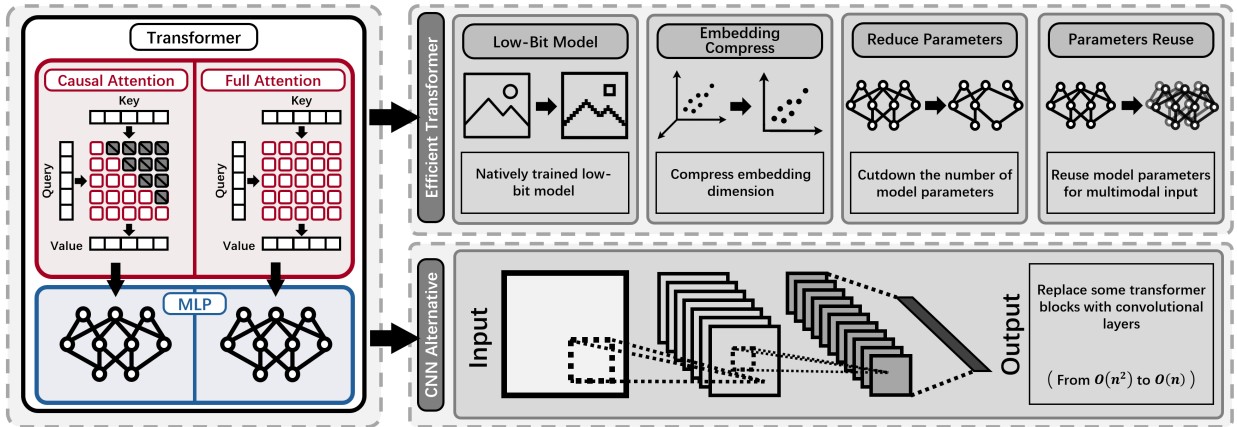

Figure 5: Overview of efficient architecture methodologies. These strategies diverge from the standard Transformer (left), branching into 'Efficient Transformer' approaches (top right) such as low-bit quantization, embedding compression, and parameter reduction/reuse, and 'CNN Alternatives' (bottom right) which replace attention blocks with convolutional layers to achieve linear $O(n)$ scaling complexity.

and energy costs, particularly during large-scale training and deployment (Vaswani et al., 2017; Rives et al., 2021; Turnbull et al., 2024; Lu et al., 2025; Yang et al., 2024).

- **Redundancy and Massive Activations in Embeddings:** Protein embeddings in large models are highly compressible, indicating over-parameterization, and also exhibit massive activations in certain channels, leading to inefficient memory use (Lu et al., 2025).

- **Limited Inductive Bias and Information Integration:** Existing architectures may not efficiently leverage important biological information, such as evolutionary relationships or sequence homology, constraining their performance and flexibility in specialized tasks (Sgarbossa & Bitbol, 2025).

The following section reviews recent architectural strategies developed to overcome these challenges. Table 2 provides an overview of these methods, and Figure 5 illustrates representative efficient architectural designs.

## 3.2 Efficient Transformer

**2Bits of Protein** (Turnbull et al., 2024) is a ternary-quantization architecture (based on (Courbariaux et al., 2016)) which offers an efficient solution to the computational and memory bottlenecks of large pLMs. In this approach, all linear layers except the embedding and output head are quantized to ternary precision, with weights constrained to $\{-1, 0, 1\}$ by scaling and rounding each weight matrix (Ma et al., 2024; Li et al., 2016; Zhu et al., 2016). Specifically, for a weight matrix $W$, the quantized matrix $\widetilde{W}$ is defined as:

$$\widetilde{W} = \text{RoundClip}\left(\frac{W}{\gamma + \epsilon}, -1, 1\right),$$

where $\gamma$ is the mean absolute value of $W$'s elements, and $\text{RoundClip}(x, a, b)$ rounds $x$ to the nearest integer within $[a, b]$.

This design achieves substantial reductions in GPU memory and energy usage, prior NLP work reports up to $3.55\times$ lower memory and $21.7\times$ less energy consumption compared to full-precision models (Ma et al., 2024). The ternary pLM preserves the standard transformer encoder architecture (Vaswani et al., 2017; Lin et al., 2023) and remains trainable from scratch with stable performance (Banner et al., 2019). On ProteinGym (Notin et al., 2022), ternary ESM2-8M attains a mean spearman-correlation of 0.181 across 217 zero-shot tasks, competitive with efficient baselines such as ProtGPT2 (Ferruz et al., 2022) and UniRep (Alley et al., 2019). Under matched compute (24 GPU-hours, 'crammed' training (Frey et al., 2024)), the

performance gap becomes statistically insignificant, whereas under the standard training schedule, ESM2-8M is about 25% higher and statistically significant ($p < 0.05$). Thus, ternary-quantization provides a principled and highly efficient design for pLMs, substantially reducing computational cost while maintaining competitive predictive performance.

**Compressed Hourglass Embedding Adaptations of Proteins (CHEAP)** (Lu et al., 2025) provide an architectural solution for drastically reducing the dimensionality and memory footprint of pLMs embeddings, while retaining both sequence and structure information. CHEAP employs a transformer-based hourglass-autoencoder (Newell et al., 2016; Kingma & Welling, 2013), which applies linear downsampling and projection to the ESMFold (Lin et al., 2023) latent embedding. Its key innovations can be divided into:

- *Continuous Compression:* CHEAP enables up to 128× channel compression and 8× sequence length compression through a learnable bottleneck, allowing backbone structure reconstruction with less than 1.34 Å root-mean-squared distance (Kabsch, 1976) at 32× channel compression, and maintaining sequence recovery above 99% until fewer than 8 channels remain. To address abnormal massive activations in certain embedding channels, CHEAP introduces per-channel normalization (Ba et al., 2016; Ioffe & Szegedy, 2015), further improving compressibility.

- *Discrete (Tokenized) Compression:* CHEAP supports discrete embeddings via finite scalar quantization (Mentzer et al., 2024), which outperforms VQ-VAE (Van Gerven & Bohte, 2017; Van Den Oord et al., 2017) for large codebooks on latent and structure reconstruction, and it also provides a representation of protein structure obtainable from sequence alone.

These compressed embeddings enable efficient downstream applications, such as protein design, similarity search, and functional prediction, making advanced pLM representations accessible for resource-limited scenarios while preserving high-resolution information.

**RAG-ESM** (Sgarbossa & Bitbol, 2025) introduces a parameter-efficient encoder-decoder architecture that augments pretrained ESM2 (Lin et al., 2023) with a minimal number of cross-attention layers and extensive parameter-sharing. The encoder embeds a context (homologous) sequence using the original ESM2 model, while the decoder reuses pretrained ESM2 layers, inserting a few cross-attention layers to transfer information from the context embedding to the masked input sequence. Critically, self-attention and feed-forward layers in both modules share weights, so the number of new parameters is negligible compared to the backbone model. Formally, the total parameter count of RAG-ESM is

$$N_{\text{RAG-ESM}} = N + M \quad (M \ll N),$$

where $N$ is the number of parameters in the pretrained ESM2 backbone, and $M$ is the number of newly added cross-attention parameters. Conditioning on homologs during inference allows RAG-ESM to extract evolutionary information (Altschul et al., 1990) without requiring MSAs (Jumper et al., 2021; Remmert et al., 2012; Rao et al., 2021) or significantly increasing model size.

Empirically, RAG-ESM models with 12M and 165M parameters achieve 48% and 43% lower perplexity, respectively, than their ESM2-8M and ESM2-150M baselines on masked AA prediction. Both are trained under the discrete-diffusion (Austin et al., 2021; Hoogeboom et al., 2021) using the closest homolog as context (Steinegger & Söding, 2017), with modest compute budgets (50–120 GPU-hours). Cross-attention heads naturally learn alignment (Vig & Belinkov, 2019); several heads reach $\rho > 0.6$, and a logistic regression trained on all 60 heads achieves an average $\rho = 0.76$ against Needleman–Wunsch (Needleman & Wunsch, 1970) alignment matrices.

This design enables efficient and scalable homology-aware pLMs that achieve state-of-the-art performance among sequence-based models for homolog-conditioned generation and remain competitive on motif scaffolding (Dauparas et al., 2022), while retaining flexibility.

**Cramming Protein Language Model Training in 24 GPU Hours** (Frey et al., 2024) introduces an efficient transformer-based architecture specifically optimized for rapid, resource-constrained pre-training. The corresponding pre-training setup is detailed in Section 4.2.

The core approach adapts the HuggingFace ESM2 (Lin et al., 2023) backbone by removing: (1) *all query, key, and value biases in attention blocks*, (2) *biases in intermediate linear layers*, thereby reducing computational

overhead while retaining model capacity to maximize per-token throughput under a stringent compute budget: model weights are initialized from scratch, and training is limited within 24 hours on a single GPU.

These architectural and implementation choices collectively allow a 67M-parameter model to achieve competitive downstream performance, on tasks such as protein fitness landscape inference (Notin et al., 2023) and protein–protein interaction classification (Szklarczyk et al., 2021), with models trained for over $15,000\times$ greater GPU hours. It demonstrates that by simplifying architectural details, it is feasible to obtain expressive pLMs suitable for practical research and deployment under extreme computational constraints.

### 3.3 Convolutional Layer Based Methods

**Convolutional Autoencoding Representations of Proteins (CARP)** (Yang et al., 2024) demonstrates that convolutional architectures can match or exceed transformer-based pLMs (Rives et al., 2021; Lin et al., 2023) in both predictive power and computational efficiency. CARP replaces transformer self-attention with ByteNet-style (Kalchbrenner et al., 2016) dilated convolutional blocks (Yu & Koltun, 2015; Chen et al., 2017), exponentially expanding the receptive-field while ensuring linear scaling in computation and memory with sequence length. Formally, the computational complexity per layer is:

$$\begin{cases} \mathcal{O}(L^2) & \text{for transformer self-attention} \\ \mathcal{O}(L) & \textbf{for convolution in CARP} \end{cases}$$

where $L$ is the input sequence length.

CARP-640M (640M parameters) achieves a masked language modeling loss of 2.02, close to ESM-1b's (Rives et al., 2021) 1.96, and attains higher average zero-shot mutation effect prediction (Spearman 0.49 vs. 0.46) on 41 datasets. Unlike ESM-1b, CARP processes sequences up to 4,096 residues without memory overflow and maintains stable loss and runtime for longer sequences. These results support convolutional architectures as efficient, scalable, and competitive alternatives to transformers for large protein sequence modeling.

**ProtHyena** (Zhang & Okumura, 2024) introduces the Hyena operator (Poli et al., 2023) as an efficient alternative to self-attention and standard convolution in pLMs. The per-layer computational complexity is:

$$\begin{cases} \mathcal{O}(L^2) & \text{Transformer self-attention} \\ \mathcal{O}(L) & \text{Standard CNN} \\ \mathcal{O}(L\log_2 L) & \textbf{The Hyena operator} \end{cases}$$

where $L$ is the input sequence length.

Compared to transformer models whose self-attention layers require quadratic complexity, the Hyena operator enables substantially improved scalability; compared to standard CNNs, whose long-range modeling is limited by kernel size, Hyena leverages implicit long convolutions (Gu et al., 2021) and element-wise gating (Dauphin et al., 2017; Hochreiter & Schmidhuber, 1997) to efficiently capture both local and global dependencies. The core block applies:

$$y = x_N \cdot (h_N * (x_{N-1} \cdot (h_{N-1} * \cdots x_1 \cdot (h_1 * v))))$$

where $N$ is block depth, $h_i$ are long convolution filters, $v$ is the input projection, and $\cdot$ denotes gating.

This enables ProtHyena to process sequences up to a million residues and achieve accuracy competitive with or better than larger transformer-based pLMs, such as ESM-1b (Rives et al., 2021), ESM-2 (Lin et al., 2023), and ProteinBERT (Brandes et al., 2022), using about 10% of their parameters and up to $60\times$ speedup.

> **Takeaway in §3 for Efficiency-focused pLMs Model Architectures**
>
> Recent architectural advances, including quantization, embedding compression, cross-attention with homology conditioning, convolutional alternatives, and the design of transformer architectures optimized for rapid "cramming" pre-training, directly address key computational and representational bottlenecks in pLMs, enabling efficient and scalable model design without sacrificing predictive power. Proteins are simultaneously more compressible and longer than natural language text, making architectural efficiency both more achievable and more necessary for pLMs. The small amino acid vocabulary, highly compressible embeddings, and strong local sequential patterns mean that aggressive techniques, ternary quantization, $128\times$ embedding compression, replacing attention with convolutions, retain more biological signal than equivalent compression would in NLP. At the same time, protein sequences reaching millions of residues make sub-quadratic architectures not merely beneficial but essential, a length regime rarely encountered in natural language.

## 4 Efficient Training

### 4.1 Background and Challenges

Training LLMs generally follows two stages: *pre-training* and *post-training.* During pre-training, models are trained on large unlabeled corpora to learn general representations (Radford et al., 2018; Devlin et al., 2019). post-training then adapts these models to downstream objectives (including, for example, supervised fine-tuning (SFT) (Raffel et al., 2020) and reinforcement learning (RL)-based tuning such as RLHF (Ouyang et al., 2022) or GRPO (Shao et al., 2024)). In this paper, we mainly focus on *fine-tuning* (Howard & Ruder, 2018) within post-training. In parallel, *distillation* transfers knowledge from high-capacity teachers to compact students for efficiency (Hinton et al., 2015; Gou et al., 2021; Sanh et al., 2019).

Training pLMs adopt a conceptually similar pipeline to LLMs but with several domain-specific characteristics: (1) *Pre-training:* pLMs are trained on large corpora of unlabeled protein sequences at the amino-acid level to learn general biological representations (Elnaggar et al., 2021; Rives et al., 2021; Bepler & Berger, 2019; Alley et al., 2019). (2) *Post-training:* pLMs are further adapted on task-specific datasets for specialized bioinformatics objectives (Min et al., 2017), for example, function prediction (Schmirler et al., 2024; Gligorijević et al., 2021). (3) *Distillation:* knowledge is transferred from large teacher models to smaller students for efficient deployment, which closely mirrors the approach used in LLMs (Shang et al., 2024; Hu et al., 2024; Sanh et al., 2019; Sun et al., 2020).

Given these characteristics, improving computational and data efficiency has become a key objective in pLMs training. Efficiency-focused pLMs training therefore focuses on the following bottlenecks:

- *Challenges in Pre-training:*

  - **Lack of Resource Allocation Guidelines:** It is challenging to optimally balance model size and training dataset size under fixed compute, with suboptimal allocation leading to overfitting (Srivastava et al., 2014) or diminished returns (Cheng et al., 2024).
  - **Limited Structural Integration:** Most pre-training pipelines rely solely on sequence data, whereas structural information remains underutilized, which may ultimately limit both the efficiency and generalization capabilities of the models (Heinzinger et al., 2024).

- *Challenges in Fine-tuning:*

  - **Lack of Comprehensive Empirical Studies:** Compared to NLP (Ding et al., 2022), systematic evaluations of parameter-efficient fine-tuning (PEFT) (Houlsby et al., 2019; Hu et al., 2022; Lester et al., 2021; Li & Liang, 2021)methods for pLMs across diverse tasks has been limited, leaving practical guidelines unclear (Schmirler et al., 2024).
  - **High Computational Demands:** Full-model fine-tuning of large pLMs requires substantial memory and compute, limiting accessibility and scalability, especially under limited hardware situations (Schmirler et al., 2024; Clancy et al., 2025).

Table 3: Summary of representative training-stage efficiency methods for pLMs.

| Category | Method | Brief Description | Limitation |
|---|---|---|---|
| Pre-training | Scaling Law (Cheng et al., 2024) | Empirical study on pLMs optimal data/model allocation. | Diminishing returns due to data redundancy, high computational costs for MLM, and sublinear data scaling relative to model size. |
| | Structure Integration (Heinzinger et al., 2024) | Joint sequence-structure pre-training (e.g., ProstT5). | Data bias that favors high-confidence, well-structured proteins over disordered ones, & potential catastrophic forgetting of sequence-only knowledge. |
| | Cramming Challenge (Frey et al., 2024) | Train a performant pLM in 24 hours on a single GPU. | Constraints that favor per-token efficiency over raw scaling, on-the-fly data preparation, restricted model sizes, inability to use pre-trained weights. |
| Fine-tuning | A study of PEFT Methods (Schmirler et al., 2024) | LoRA for parameter-efficient tuning. | Optimal PEFT choice is task- and model-dependent. Often leaves a small performance gap compared to full-parameter fine-tuning |
| | QLoRA (Clancy et al., 2025) | 4-bit quantization + LoRA for efficient fine-tuning. | Gap between full fine-tuning and QLoRA tends to widen as the model size increases, activation memory uses lots of VRAM, require lower learning rates to remain stable |
| | SES-Adapter (Tan et al., 2024) | Structure-aware adapters for cross-modal representation. | Reduced predictive accuracy compared to full tuning, high sensitivity to structural noise from predicted models, and significant computational overhead during inference. |
| | FSFP (Zhou et al., 2024) | few-shot with meta-learning and LoRA. | Meta-learning overhead; assumes related auxiliary tasks exist, potential overfitting to the support set in low data. |
| Distillation | Cross-distillation in AlphaFold3 (Abramson et al., 2024) | Distillation from a single large teacher-model. | Risk of inherited teacher biases, potential structural stereotypy (loss of conformational diversity), and computational costs for generating high-quality teacher structures. |
| | ProtGO (Hu et al., 2024) | Transferring functional and structural knowledge. | Reliance on limited functional annotations, dependency on external pre-training for performance, and the need for significant computational resources to align multiple modalities. |
| | VespaG (Marquet et al., 2024) | Distillation via evolutionary expert supervision. | Performance decline when predicting effects on de novo designed proteins compared to natural ones. |
| | MTDP (Shang et al., 2024) | Aggregating knowledge from multiple teacher models. | Restricted training data scope (UniProtKB), a 1,000-residue sequence length cap, and potential conflicts between teacher models. |

  – **Low Data Efficiency:** Adapting models to new tasks with limited labeled data can be inefficient, as conventional fine-tuning is prone to overfitting and resource waste (Zhou et al., 2024).

- *Challenges in Distillation:*

  – **Teacher-student Discrepancy:** Differences in size, architecture, or modality (e.g., sequence, structure, function) between teacher and student models can hinder efficient knowledge transfer and limit student performance (Abramson et al., 2024; Hu et al., 2024; Mirzadeh et al., 2020).
  – **Knowledge Transfer Bottlenecks:** Distilling large teacher models or integrating multi-modal knowledge (e.g., structure, function) into compact students can reduce predictive power or require extensive tuning (Marquet et al., 2024; Hu et al., 2024; Shang et al., 2024).

Following sections review methods addressing challenges through efficient pre-training, fine-tuning, and knowledge distillation methods.

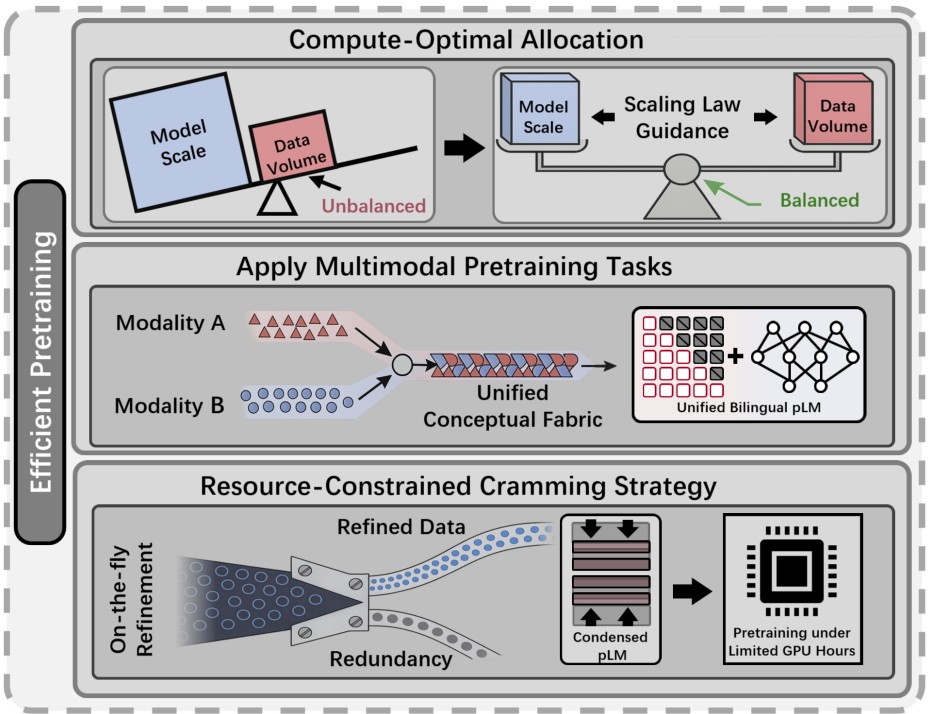

Figure 6: Conceptual illustration of efficient pretraining strategies: Compute-Optimal Allocation (top) ensures the balanced scaling of model parameters and data volume guided by established scaling laws; Apply Multimodal Pretraining Tasks (middle) integrates distinct biological modalities into a unified conceptual fabric to enhance representation; and Resource-Constrained Cramming Strategy (bottom) leverages on-the-fly data refinement and condensed architectures to achieve high performance within limited GPU hours.

## 4.2 Efficient Pre-training Methods

To orient the discussion, Figure 6 summarizes the efficient pre-training methods reviewed in this section.

**Scaling law-guided pre-training for pLMs** (Cheng et al., 2024) empirically establishes optimal resource allocation between model size and data for transformer-based pLMs. On a large dataset of 939M protein sequences, over 300 models were trained to reveal that both masked (MLM) and causal (CLM) objectives follow distinct power-law scaling similar to Kaplan et al. (2020) and Hestness et al. (2017). For a $10\times$ increase in compute, optimal MLM size increases $6\times$ with $1.7\times$ more data, while CLM size grows $4\times$ with $3\times$ more data. The fitted scaling relations for MLMs are:

$$\begin{cases} N_{\mathrm{MLM}}^{\mathrm{opt}} = 6.19 \times 10^{-8} C^{0.776} & \text{MLMs' model parameters} \\ D_{\mathrm{MLM}}^{\mathrm{opt}} = 2.02 \times 10^{6} C^{0.230} & \text{MLMs' training tokens} \end{cases}$$

and for CLMs are:

$$\begin{cases} N_{\mathrm{CLM}}^{\mathrm{opt}} = 1.26 \times 10^{-3} C^{0.578} & \text{CLMs' model parameters} \\ D_{\mathrm{CLM}}^{\mathrm{opt}} = 1.23 \times 10^{2} C^{0.422} & \text{CLMs' training tokens} \end{cases}$$

where $C$ represents the total pre-training FLOPs, N is the number of forward-activated non-embedding parameters, and D represents the number of training tokens.

Repeating data leads to overfitting for MLMs and diminishing returns for CLMs (Hernandez et al., 2022), which is mitigated by introducing a diverse, deduplicated dataset (UniMeta200B). Transfer learning, pre-training with CLM and then fine-tuning with MLM, reduces the compute needed for optimal MLMs by up to $7.7\times$. Large-scale validations confirm that these scaling laws yield models with better generalization and

downstream performance under fixed compute budgets, particularly in resource-constrained environments.

**ProstT5** (Heinzinger et al., 2024) is a T5-based model (Raffel et al., 2020) that integrates structure information (Jing et al., 2020) into pLM pretraining to boost efficiency and versatility. ProstT5 unifies protein sequence (Amino Acid, AA) and structural (3Di token) modalities within a single encoder-decoder architecture. Its key innovations in the pretraining stage can be divided into:

- *Multimodal Pre-training:* By encoding 3D structures from AlphaFoldDB (Varadi et al., 2024) as 3Di-token sequences, and expanding the vocabulary with 3Di tokens, ProstT5 applies span-based denoising objectives to both AA and 3Di sequences, teaching the model new structural tokens while avoiding catastrophic forgetting of sequence information.

- *Bi-directional Translation Objectives:* ProstT5 is trained to translate between AA and 3Di representations using direction tags (`<fold2AA>`, `<AA2fold>`), enabling both "folding" (AA→3Di) and "inverse folding" (3Di→AA), thus robustly linking sequence and structure information.

This bilingual setup allows ProstT5-predicted 3Di strings to be used for structure-level similarity search (e.g., with Foldseek (Van Kempen et al., 2024)), achieving remote homology detection (Rost, 1999) sensitivity nearly matching experimental structures, but with a speedup of over three orders of magnitude (e.g., 43 seconds vs. 48 hours for full proteome annotation) (Van Kempen et al., 2024; Mirdita et al., 2022). Embedding-based annotation transfer further shows improved CATH fold classification (Sillitoe et al., 2021) accuracy over ProtT5 (Elnaggar et al., 2021) and ESM-1b (Rives et al., 2021), and secondary structure prediction with ProstT5 matches or exceeds prior state-of-the-art. In inverse folding (Dauparas et al., 2022; Hsu et al., 2022), ProstT5 generates diverse sequences with predicted structures closely matching targets, demonstrating both enhanced efficiency and versatility.

Overall, integrating structure into pLM pretraining via bilingual modeling delivers substantial gains in inference speed and structural generalization, advancing protein design and large-scale annotation capabilities.

**Cramming Protein Language Model Training in 24 GPU Hours** (Frey et al., 2024) proposes a pre-training paradigm specifically tailored for rapid and resource-constrained scenarios. In this framework, a transformer-based pLM is trained from scratch under a strict 24 GPU-hour budget, with no use of pre-trained models at any stage. The pre-training pipeline consists of several key components:

- *Random initialization:* All model weights are initialized randomly.

- *On-the-fly masked language modeling:* Masked language modeling is performed directly on UniRef50 (Suzek et al., 2015) splits, with all data processing (e.g., tokenization, filtering, sorting) occurring on-the-fly during training, and no offline pre-processing.

- *Large effective batch size:* Training employs a batch size of 128, sequence length of 512, and gradients are accumulated over 16 steps for an effective batch of 2048 sequences.

- *Critical hyperparameter tuning:* The optimizer is AdamW (Loshchilov & Hutter, 2017) with carefully tuned parameters. The learning rate schedule is central: the optimal regime employs a fast warmup (Goyal et al., 2017; Liu et al., 2020) to a peak learning rate of $1 \times 10^{-3}$ over 1,000 steps, followed by a slow linear decay (Loshchilov & Hutter, 2016) to near zero over the remaining steps, with a training process capped at 50,000 updates.

These pre-training choices allow the resulting 67M-parameter model to match or approach the downstream performance of much larger models such as ESM2-3B (Lin et al., 2023), trained for over 15,000× more GPU hours, on tasks including protein fitness landscape inference and PPI classification. The architectural design supporting these results is detailed in Section 3.2. This work demonstrates that, through systematic pre-training design and hyperparameter tuning (Bergstra & Bengio, 2012; Snoek et al., 2012), performant pLMs can be obtained at a fraction of the conventional computational cost, enabling broader accessibility for rapid biological modeling.

**Takeaway in §4.2 for Efficiency-focused pLM Pre-Training**

Scaling law-guided resource allocation, the integration of structural information, and the development of rapid "cramming" pre-training protocols have substantially improved the efficiency and versatility of pLMs pre-training, providing practical solutions for balancing compute, data, and biological knowledge, even under stringent resource constraints, in large-scale protein modeling. Pre-training efficiency for pLMs is shaped by two properties absent in NLP. First, protein databases are structured by evolutionary homology, making data diversity, not volume, the binding constraint: family-level redundancy penalizes naïve repetition, while diverse metagenomic sources and cross-objective transfer deliver outsized returns. Second, protein structure is computationally derivable at scale from sequence alone, providing a second pre-training modality at near-zero marginal cost, enabling bilingual approaches like ProstT5 to achieve structural generalization that sequence-only scaling cannot replicate.

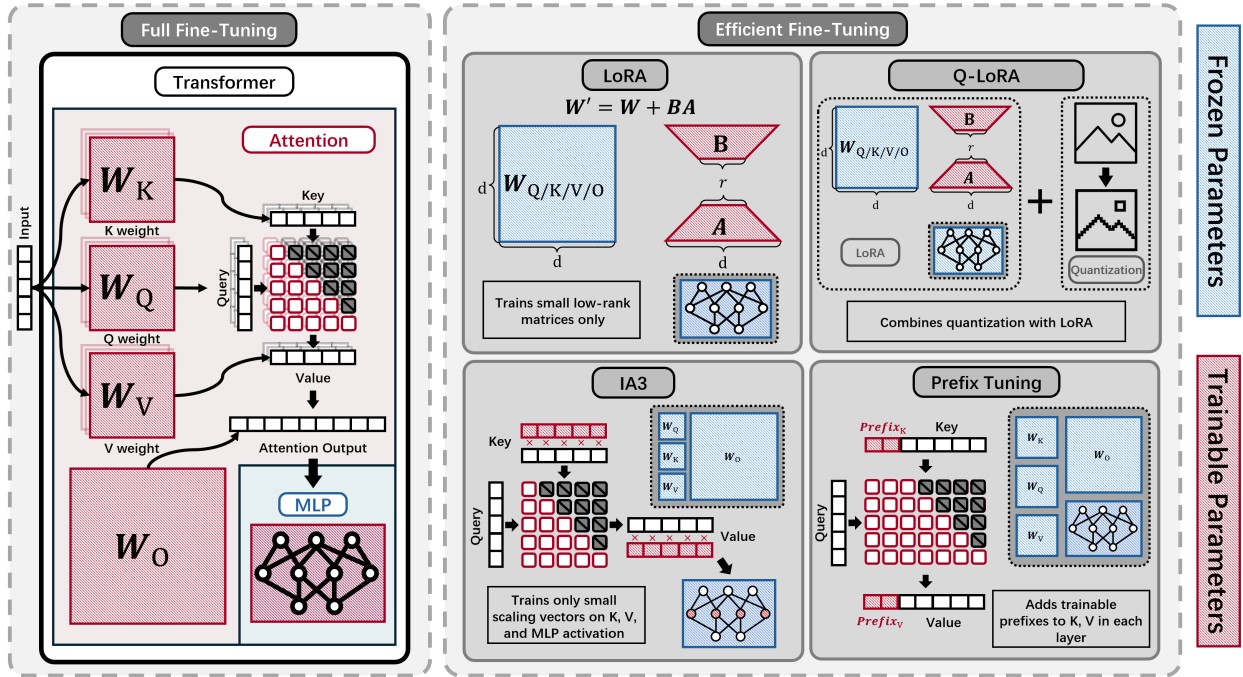

Figure 7: A comparison of Full Fine-Tuning (left) with several mainstream Parameter-Efficient Fine-Tuning (PEFT) methods (right) for pLMs. In full fine-tuning, all model parameters are trainable. PEFT methods keep the backbone parameters frozen (blue) and introduce a small number of new trainable parameters (pink). Examples shown include LoRA, which adds low-rank adaptation matrices; Q-LoRA, which combines LoRA with quantization; IA3, which trains scaling vectors; and Prefix Tuning, which adds trainable prefixes to key and value vectors.

## 4.3 Efficient Fine-tuning Methods

Fine-tuning in pLMs involves adapting a pre-trained model to specific downstream biological tasks using relatively small amounts of labeled protein data (Schmirler et al., 2024). This step is inherently efficiency-focused, enabling rapid model adaptation without extensive retraining. Advanced PEFT techniques, such as LoRA (Hu et al., 2022) and structure-aware adapters (Tan et al., 2024), further minimize computational and memory demands while maintaining high performance. Subsequent subsections review these PEFT methods and associated efficiency gains.

A comparison of full fine-tuning with PEFT methods is illustrated in Figure 7.

**A study of Parameter-efficient fine-tuning (PEFT) methods** systematically compared full-model fine-tuning with several PEFT approaches, including LoRA (Hu et al., 2022), DoRA (Liu et al., 2024a), IA3 (Liu et al., 2022), and prefix tuning (Li & Liang, 2021), across diverse pLMs and protein prediction tasks (Schmirler et al., 2024). Its main findings include:

- *Efficiency and accuracy:* By freezing most model weights and only updating a small fraction (e.g., 0.25% for LoRA), PEFT methods, such as LoRA, achieved nearly equivalent accuracy to full fine-tuning.

- *Training speedup and memory savings:* For larger models, LoRA offered up to a 4.5× training speedup with comparable GPU memory requirements, and saving only adapter weights, further improving efficiency.

- *Compatibility and resource efficiency:* LoRA, DoRA, and similar methods are compatible with mixed precision, gradient accumulation, and CPU-offloading, enabling fine-tuning even on 8GB or 16GB GPUs.

- *Method comparison:* Among PEFT methods, LoRA was generally most compute-efficient for large pLMs, with DoRA sometimes up to 30% slower, and all PEFT methods yielded average prediction gains of 61.3% compared to using static embeddings.

PEFT, especially LoRA, provides a practical solution for adapting large pLMs in resource-constrained environments, with all tested PEFT methods demonstrating substantial improvements over static, pre-trained embeddings.

**Quantization with LoRA (QLoRA)** (Clancy et al., 2025) combines 4-bit quantization with LoRA to enable highly efficient fine-tuning of pLMs while maintaining strong performance (Dettmers et al., 2023). By quantizing most model weights to 4 bits and updating only a small set of adapter parameters, QLoRA substantially reduces memory and computation costs.

Clancy et al. (Clancy et al., 2025) systematically evaluated QLoRA across multiple pLMs, including ESM-2 (Lin et al., 2023), ESM C (Team, 2024), ProtBERT (Brandes et al., 2022), ProtT5-half (Elnaggar et al., 2021), and Ankh-base (Elnaggar et al., 2023), and diverse tasks, including GFP fluorescence (Sarkisyan et al., 2016), protease stability (Rocklin et al., 2017), and protein secondary structure prediction (Berman et al., 2000; Kryshtafovych et al., 2021; Klausen et al., 2019). Experiments show that QLoRA achieves an average GPU memory reduction of 46.7%, with generative models such as ProtGPT2 and ProLLaMA reaching up to 76.4% reduction.

Performance remains largely preserved: for ESM C 600M, fluorescence prediction SpearmanR is 0.850 for QLoRA versus 0.863 for full precision, and secondary structure accuracy remains identical at 0.870. In generative tasks, 4-bit quantization yields negligible differences in predicted protein quality, with metrics such as Foldseek pass rate, local distance difference test, and predicted local distance difference test showing no significant decline.

Overall, QLoRA lowers hardware barriers for fine-tuning and inference of large pLMs, making efficient protein modeling feasible on modest computing infrastructure.

**Few-Shot Learning for Protein Fitness Prediction (FSFP)** (Zhou et al., 2024) builds on the dataset construction and meta-learning strategies described in Section 2.2, and focuses on PEFT for robust fitness prediction with minimal labeled data.

After meta-training on auxiliary tasks constructed from related proteins and MSA-based pseudo-labels, FSFP fine-tunes only a small subset of parameters (e.g., via LoRA with rank $r = 16$) to mitigate overfitting. Fitness prediction is formalized as a ranking problem, with ListMLE loss (Xia et al., 2008) used to optimize the ordering of variants. On 87 deep mutational scanning datasets, FSFP outperforms both zero-shot and regression-based baselines, improves spearman-correlation by up to 0.1 with just 20 labeled mutants, and demonstrates strong extrapolation to unseen mutations. Wet-lab validation further highlights its efficiency and real-world utility.

**SES-Adapter** (Tan et al., 2024) is a structure-aware, PEFT framework for pLMs, integrating sequence and structural information via cross-modal multi-head attention. In SES-Adapter, the pLM backbone is frozen, and only the adapter module is trained. Structural features representations are serialized via, such

as, FoldSeek (Van Kempen et al., 2024) or DSSP (Kabsch & Sander, 1983), into embeddings and serve as queries ($Q$), while pLM sequence embeddings serve as keys and values ($K$, $V$). Uniquely, both $Q$ and $K$ are first embedded with rotary position embedding (RoPE). For an embedding vector $x \in \mathbb{R}^{d_k}$ at position $p$, RoPE performs a rotation on each coordinate pair $(2i, 2i+1)$:

$$\begin{bmatrix} (\text{RoPE}(x,p))_{2i} \\ (\text{RoPE}(x,p))_{2i+1} \end{bmatrix} = \begin{bmatrix} \cos\theta_{p,i} & -\sin\theta_{p,i} \\ \sin\theta_{p,i} & \cos\theta_{p,i} \end{bmatrix} \begin{bmatrix} x_{2i} \\ x_{2i+1} \end{bmatrix}, \quad i = 0, \ldots, \frac{d_k}{2} - 1,$$

where $\theta_{p,i}$ is a position-dependent rotation angle varying with $p$ and the dimension index $i$.

This fusion enables SES-Adapter to consistently outperform vanilla pLMs and prior tuning strategies: on nine diverse benchmarks, it yields up to 11% (avg. 3%) accuracy improvement, convergence acceleration up to 1034% (avg. 362%), and maintains robustness to structural noise (performance variation <0.6% across different structure predictors). SES-Adapter thus offers an efficient, scalable, and model-agnostic solution for fine-tuning pLMs in protein research.

---

**Takeaway in §4.3 for Efficiency-focused pLMs Fine-Tuning**

Recent advances in PEFT, such as LoRA, QLoRA, meta-learning, and structure-aware adapters, enable large pLMs to be effectively adapted with limited computational resources and small labeled datasets. These approaches address key bottlenecks in scalability and accessibility, and provide practical solutions for efficient, robust downstream adaptation in diverse biological tasks. Protein fine-tuning faces an extreme version of NLP's adaptation challenge: labeled data requires costly wet-lab experiments (often yielding only tens of examples), and many biology labs lack data-center-scale compute. Two protein-specific priors make PEFT viable despite these constraints. Evolutionary relatedness across protein families enables meta-learning transfer from homologous datasets, allowing methods like FSFP to succeed with as few as 20 labeled mutants. Meanwhile, 3D structure, computationally derivable from sequence at zero annotation cost, provides a complementary fine-tuning signal (SES-Adapter) unavailable in text domains.

---

### 4.4 Efficient Training via Distillation

knowledge distillation (Hinton et al., 2015) in pLMs is a widely adopted model compression technique that transfers knowledge from larger, high-capacity teacher models to smaller, efficient student models. It typically involves guiding student models to mimic teacher outputs, intermediate representations, or other informative signals(Zagoruyko & Komodakis, 2016). Distillation inherently emphasizes efficiency, enabling deployment of lightweight models that substantially reduce computational and memory requirements (Cheng et al., 2017; Han et al., 2015) while retaining predictive capabilities comparable to larger models (Wang & Yoon, 2021; Geffen et al., 2022). Recent literature highlights diverse distillation strategies, including single-teacher (Abramson et al., 2024), expert-guided (Marquet et al., 2024), cross-modal (Hu et al., 2024), and multi-teacher adaptive methods (Shang et al., 2024), demonstrating significant efficiency improvements.

**Cross-distillation of AlphaFold 3** (Abramson et al., 2024) is a strategy for mitigating hallucinations and improving data efficiency in diffusion-based protein structure prediction (Watson et al., 2023). Its core methodological steps are:

- *Teacher-augmented training data:* The training set is augmented by including high-confidence structures predicted by AlphaFold-Multimer v2.3 (Evans et al., 2021; Židek, 2022), in addition to experimental structures. Teacher-model predictions act as additional supervisory targets during training (Lee, 2013; Xie et al., 2020).

- *Joint supervision with ground-truth and teacher outputs:* AlphaFold 3 is explicitly optimized to align its outputs not only with experimental ground-truth structures, but also with reliable teacher-generated structures. This enables the model to learn more physically realistic disorder regions, as teacher-generated structures tend to represent unstructured segments as extended loops, reducing hallucinated compact order in intrinsically disordered regions.

On the CAID 2 benchmark (Kryshtafovych et al., 2021), cross-distillation led to better alignment between predicted confidence (predicted local distance difference test) and actual disorder, effectively suppressing generative artifacts. By increasing both the diversity and quantity of usable training data, intra-chain metrics reach 97% of their maximum within the first 20,000 training steps, reflecting rapid and efficient learning. Overall, cross-distillation exemplifies how leveraging both ground-truth and high-quality model-generated supervision can simultaneously reduce hallucinations and accelerate robust protein structure learning in next-generation diffusion-based models.

**ProtGO** (Hu et al., 2024) is a multimodal protein representation learning framework that distills functional knowledge from a teacher to a student network via distribution alignment in latent space. The teacher leverages sequence, structure, and GO (GO) (Ashburner et al., 2000; Aleksander et al., 2023) function annotations, while the student uses only sequence and structure, making it practical for annotation-scarce scenarios. Instead of aligning individual embeddings, ProtGO minimizes the KL-divergence (Kullback & Leibler, 1951) between the batch-level Gaussian-distributed graph embeddings of teacher and student:

$$L_{\mathrm{kd}} = \mathrm{KL}\left[p_S(z_S) \,\|\, p_T(z_T)\right]$$

where $z_S$ and $z_T$ are graph-level embeddings of the teacher and student models, and $p_S(\cdot)$, $p_T(\cdot)$ denote their estimated batch-wise distributions (Gaussian with mean and variance computed over each batch). The overall loss for training the student is:

$$L = L_{\mathrm{student}} + \beta L_{\mathrm{kd}}$$

where $L_{\mathrm{student}}$ is the cross entropy loss for downstream classification and $\beta$ is a trade-off hyperparameter.

This strategy enables the student to inherit rich functional information, achieving state-of-the-art results: e.g., 60.5% accuracy for fold classification (3.8 points higher than prior methods), 89.4% for enzyme reaction prediction, and top $F_{max}$ values on GO term and Enzyme Commission (EC) number prediction (Bairoch, 2000). Ablation studies confirm the distillation module is critical for performance (fold accuracy drops from 60.5% to 57.8% when removed). ProtGO thus demonstrates the efficiency and effectiveness of distribution-based multimodal distillation for robust protein representation learning.

**Expert-guided distillation in VespaG** (Marquet et al., 2024) offers an efficient strategy for protein variant effect prediction by training a lightweight neural network (student) to imitate an evolutionary expert model (teacher). Here, the teacher is GEMME (Laine et al., 2019), a MSA (MSA)-based method that generates mutational effect scores, while the student is VespaG, a feed-forward neural network with a single hidden layer. VespaG takes embeddings from a pretrained pLM (ESM-2 (Lin et al., 2023)) as input and learns to predict GEMME's variant effect scores for each possible mutation. The student is trained by minimizing the mean squared error between its predictions $\hat{y}$ and the GEMME scores $y^*$:

$$L = \frac{1}{N} \sum_{i=1}^{N} (\hat{y}_i - y_i^*)^2$$

This distillation approach removes the need for log-odds calculations and MSAs at inference time, enabling VespaG to rapidly predict mutational landscapes across entire proteomes on standard hardware. On the ProteinGym (Notin et al., 2022) benchmark (over 3 million variants), VespaG achieves a mean spearman-correlation of $0.48 \pm 0.02$, matching state-of-the-art methods and sometimes surpassing its expert teacher in certain protein families. Importantly, VespaG is over three orders of magnitude faster than MSA-based methods. This demonstrates that expert-guided distillation with pLM embeddings yields robust, scalable, and practical variant effect predictors.

**Multi-teacher distillation in MTDP** (Shang et al., 2024) enables efficient and high-fidelity embedding by aggregating knowledge from multiple large pre-trained models, specifically ESM2-33 and ProtT5-XL-UniRef50.MTDP student, a T5-based model with ∼20M parameters, is trained via an adaptive teacher selection mechanism that dynamically assigns the most suitable guidance to each sample using RL strategy (Yuan et al., 2021). The training objective jointly minimizes a masked language modeling loss ($L_{\mathrm{MLM}}$) and

a distillation loss ($L_{\text{Distill}}$) defined by the KL divergence between student and teacher output distributions:

$$L = \alpha L_{\text{MLM}} + (1 - \alpha) L_{\text{Distill}}, \quad \alpha = 0.2$$

where,

$$L_{\text{MLM}} = -\sum_{i=1}^{N} \sum_{j=1}^{V} I(y_i = j) \log p(x_i = j)$$

$$L_{\text{Distill}} = \sum_{i=1}^{N} \text{KL}\left(p_T(x_i) \parallel p_S(x_i)\right)$$

where $N$ is sequence length, $V$ is vocabulary size, $y_i/x_i$ are the true/predicted amino acids at position $i$, $I(\cdot)$ is the indicator function, $p_T(x_i)/p_S(x_i)$ are the teacher/student output probabilities, and KL denotes KL divergence. At each step, selected teacher $T$ for each sample is chosen via a RL-based scheduling mechanism (Yuan et al., 2021).

Compared to its teachers (650M and 120M parameters), the MTDP student is dramatically more efficient, achieving $\pm1.5\%$ accuracy of large models on tasks like function and subcellular localization prediction, and requiring $\sim70\%$ less encoding time for 10,000 protein sequences. Ablation studies confirm multi-teacher distillation outperforms single-teacher setups. MTDP can be deployed on modest GPU hardware, providing a scalable, resource-efficient solution for large-scale, real-time protein informatics applications.

---

**Takeaway in §4.4 for pLMs Distillation**

Recent progress in distillation, including expert-guided, cross-modal, and multi-teacher strategies, enables efficient pLMs with reduced memory and computational requirements. These approaches address knowledge transfer and model compression challenges, making large-scale protein analysis and deployment more practical while preserving strong predictive performance. Protein distillation differs qualitatively from its NLP counterpart: rather than simply compressing a large model into a small one, each teacher contributes a distinct biological modality that is too expensive to access at inference scale. GEMME distills MSA-derived evolutionary conservation, AlphaFold3 distills predicted 3D structure, and ProtGO distills functional annotations, all collapsed into students that require only sequence input. This transforms proteome-scale tasks (annotating $\tilde{2}00M$ sequences, predicting variant effects across entire families) from infeasible multi-modal pipelines into tractable single-forward-pass operations.

---

## 5 Efficient Inference

### 5.1 Background and Challenges

Inference in language models refers to the process of generating outputs, such as predictions or embeddings, based on trained models given new input data. Autoregressive models (e.g. GPT series (Brown et al., 2020)) generate text by predicting subsequent tokens, while masked language models (e.g. BERT (Devlin et al., 2019)) infer masked tokens or produce contextual embeddings for various tasks. This inference typically involves passing inputs through multiple transformer layers to capture contextual relationships and generate meaningful, task-relevant representations.

Analogously, pLMs infer structural, functional, or evolutionary properties from protein sequences. Given a protein sequence, pLM inference generates embeddings that encode rich biological information, facilitating downstream tasks like homolog detection, structure prediction, and functional annotation (Rives et al., 2021).

Efficiency-focused inference methods in LLMs encompass techniques such as quantization (Jacob et al., 2018; Nagel et al., 2021) and embedding-based dense retrieval, aiming to reduce computational costs and memory usage while preserving model performance (Bondarenko et al., 2021; Lewis et al., 2020). Similarly, efficient

Table 4: Summary of representative inference-stage efficiency methods for pLMs.

| Category | Method | Brief Description | Limitation |
|---|---|---|---|
| Inference | PLMSearch (Liu et al., 2024b) | Efficient remote homology search using learned embeddings. | Inability to generate sequence alignments directly, requiring supplementary tools like PLMAlign, improves remote homology detection but lacks inherent, residue-level alignment capabilities. |
| | PTQ4Protein (Peng et al., 2023) | PTQ for reducing inference memory and compute. | While 8-bit weight quantization is feasible for ProteinLMs like ESMFold, quantizing activations causes severe accuracy degradation due to wide, asymmetric distributions. |
| | DHR (Hong et al., 2024) | Transforms protein homology detection from pairwise alignment into vector retrieval. | Lacks the ability to generate explicit, residue-level alignments without integration into hybrid pipelines. |

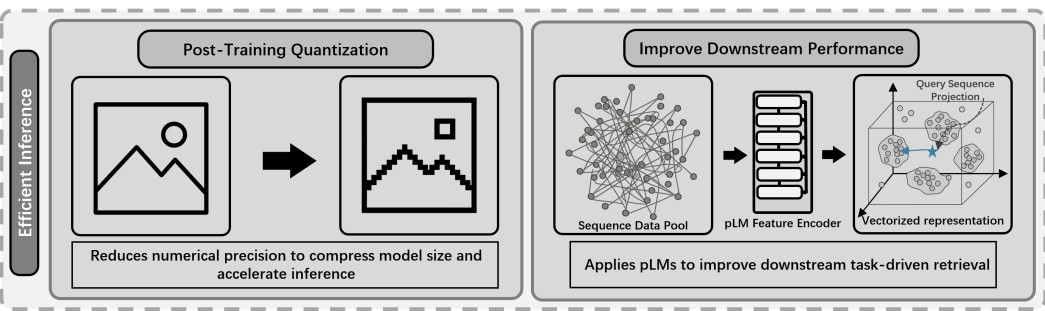

Figure 8: Strategies for improving inference efficiency: Post-Training Quantization (PTQ) (left) reduces numerical precision to compress model size and accelerate inference; Improve Downstream Performance (right) utilizes pLM feature encoders to project biological sequences into a vectorized representation, thereby optimizing downstream task-driven retrieval and similarity search.

inference in pLMs integrates analogous strategies, including PTQ (Peng et al., 2023), embedding-based retrieval via dual-encoder architectures (Hong et al., 2024), and structure-informed similarity scoring (Liu et al., 2024b). An overview of these efficient inference strategies is provided in Figure 8.

The emergence of efficiency-focused inference methods in pLMs is largely motivated by several challenges related to inference process, including:

- **Resource Constraints:** The inference process of large pLMs require substantial computation and memory, hindering deployment in resource-limited settings (Peng et al., 2023).

- **Inference Latency:** Traditional alignment-based and MSA methods are inherently slow and cannot efficiently scale to large databases, which is critical for many practical applications, such as proteome-wide homology search (Liu et al., 2024b; Hong et al., 2024).

- **Limited Sensitivity:** Sequence-based retrieval may miss remote homologs and deeper structural relationships (Liu et al., 2024b; Hong et al., 2024).

The following section reviews methods that address these issues, with an overview provided in Table 4.

## 5.2 Efficient Inference

**PTQ4Protein** (Peng et al., 2023) enables efficient inference of large pLMs such as ESMFold (Lin et al., 2023) by applying PTQ (Banner et al., 2019) to both weights and activations. Baseline experiments show that standard 8-bit uniform quantization (Jacob et al., 2018) substantially reduces memory usage (4×) but leads to significant accuracy loss, mainly due to the asymmetric and wide activation distributions before

LayerNorm layers. To address this, PTQ4Protein employs piecewise linear quantization for activations, splitting the value range $[r_l, r_u]$ at breakpoints $p_l$ and $p_u$, and quantizing each region separately:

$$\hat{r} = \begin{cases} \text{uni}(r_1; b-1, p_l, p_u, p_l) & r_1 \in [p_l, p_u] \\ \text{uni}(r_2; b-1, r_l, p_l, r_l) & r_2 \in [r_l, p_l) \\ \text{uni}(r_3; b-1, p_u, r_u, p_u) & r_3 \in (p_u, r_u] \end{cases}$$

where $[r_l, r_u]$ denotes the full activation range, $p_l$ and $p_u$ are the central breakpoints, $b$ is the total bit width, and uni($\cdot$) denotes uniform quantization within each segment. Each $r_1, r_2, r_3$ represents the activation value in the central and two tail regions, respectively.

Experiments demonstrate that PTQ4Protein achieves nearly lossless accuracy: for 8-bit quantization, TM-score (Zhang & Skolnick, 2004) drops by only 0.36% (CASP14) (Kryshtafovych et al., 2021) and 0.08% (CAMEO) (Haas et al., 2018), with memory usage reduced to 25% of baseline. Even at 6 bits, accuracy loss remains below 0.5%. Ablation studies show ESM-2 (Lin et al., 2023) is particularly robust, and the visual fidelity of predicted structures is preserved. Overall, PTQ4Protein provides a practical solution for deploying large pLMs on resource-limited hardware with minimal accuracy degradation.

**PLMSearch** (Liu et al., 2024b) is an efficient inference framework for large-scale remote homology search that leverages pLM embeddings and structure-informed similarity prediction. Its pipeline consists of:

1. *PfamClan-based pre-filtering:* PfamClan (Mistry et al., 2021) assigns proteins to evolutionarily related superfamilies (clans); this stage retains only protein pairs sharing the same clan, ensuring high recall while greatly reducing the candidate search space.

2. *SS-predictor:* A bilinear neural network is trained to predict structural similarity (TM-score (Zhang & Skolnick, 2005)) between protein pairs based on pLM embeddings; it integrates predicted TM-scores with cosine similarity, capturing global and local sequence features for remote homolog detection.

3. *PLMAlign:* For top-scoring pairs, per-residue alignment is performed using pLM embeddings to refine hit precision and enhance biological relevance.

A key efficiency innovation is the precomputation and caching of all target protein embeddings, enabling scalable, million-level searches via a single forward pass through the SS-predictor. By restricting downstream alignment to high-confidence pre-filtered pairs, PLMSearch dramatically reduces computation while maintaining high sensitivity. On standard benchmarks, PLMSearch achieves orders-of-magnitude faster search and higher sensitivity than conventional structure or sequence-based methods. These results establish PLMSearch as a scalable, structure-aware solution for accurate and ultra-fast remote protein homology inference.

**Dense Homolog Retriever (DHR)** (Hong et al., 2024) is an structure-aware inference framework for large-scale protein homolog detection, integrating pLMs with deep dense retrieval. Its workflow consists of:

1. *Dual-encoder architecture:* Both query and candidate sequences are embedded into dense vectors using a pair of ESM-initialized encoders, allowing alignment-free similarity calculation.

2. *Contrastive learning:* The dual-encoders are trained to embed homologous pairs close together and nonhomologous pairs far apart, enabling effective discrimination via dot-product scoring.

3. *Scalable dense retrieval:* All database embeddings are precomputed and cached, allowing rapid, large-scale retrieval of homologs through efficient vector similarity search (Johnson et al., 2019; Malkov & Yashunin, 2018) on standard hardware.

As an MSA prefilter, DHR accelerates MSA (with JackHMMER (Eddy, 2011)) by up to 93-fold, increases MSA diversity (log Meff (Morcos et al., 2011)), and improves AlphaFold2 (Jumper et al., 2021) structure prediction, boosting TM-score (Zhang & Skolnick, 2005) by up to 0.03 and lowering root-mean-squared distance (Kabsch, 1976) by 0.15 Å on hard targets. DHR also maintains robust performance on massive datasets such as BFD (Steinegger & Söding, 2018) or (Mitchell et al., 2020) MGnify (over 500 million sequences) (Mirdita et al., 2022), demonstrating its practicality for structure-aware, scalable, and accurate protein inference at unprecedented speed.

> **Takeaway in §5 for pLMs Efficiency-focused Infernece**
>
> Recent methods, including advanced quantization, embedding-based retrieval, and structure-informed similarity prediction, enable pLMs to achieve efficient, scalable inference with lower computational cost and memory usage, while maintaining or improving predictive performance for large-scale biological applications. Protein inference faces a challenge absent in NLP retrieval: the similarity that matters most, remote evolutionary homology, is invisible to surface-level sequence comparison, eliminating cheap baselines like keyword matching. Detecting these relationships requires either slow alignment tools (JackHMMER) or learned pLM representations, both expensive per query and infeasible when scaled to databases of 500M+ sequences. This makes precomputed embedding-based search and domain-specific quantization prerequisites for proteome-scale deployment.

## 6 Discussion

### 6.1 Intersections Across Efficiency Dimensions

The four efficiency dimensions reviewed in this survey, dataset, architecture, training, and inference, are not mutually exclusive; several methods naturally span multiple categories. We highlight these intersections to clarify the taxonomy and emphasize that the most effective efficiency strategies often combine complementary approaches. **FSFP** (Section 2.2) appears under both dataset efficiency and training efficiency (fine-tuning) because it simultaneously addresses data scarcity through meta-learning-based auxiliary task construction and reduces parameter overhead through LoRA-based adaptation. Its efficiency stems from the synergy between these two axes: meta-learning provides a better initialization that makes parameter-efficient fine-tuning more effective under few-shot conditions. **Cramming** (Frey et al., 2024) appears under both architecture efficiency (Section 3.2) and pre-training efficiency (Section 4.2) because it co-optimizes architectural choices (e.g., removing biases from attention and linear layers) with pre-training procedures (e.g., aggressive gradient accumulation, optimized learning rate schedules). Separating these contributions would obscure the key finding that architecture and training efficiency interact multiplicatively. **Scaling laws** (Cheng et al., 2024) similarly bridge dataset and training efficiency, as they govern the joint allocation of data volume and model size under fixed compute budgets.

These overlaps reflect the interconnected nature of the pLM pipeline: in practice, the largest efficiency gains are often achieved by combining techniques across dimensions, for example, using a distilled model with post-training quantization, or applying LoRA fine-tuning to an architecturally efficient model. We encourage future work to explore such combinations systematically.

### 6.2 Evaluation Benchmarks and Comparability

A challenge in comparing the methods reviewed in this survey is that different papers evaluate on different benchmarks, making direct numerical comparisons across methods difficult. The main evaluation frameworks used include:

- **Variant effect prediction:** ProteinGym (Notin et al., 2022) (used by FSFP, 2Bits of Protein, abd VespaG) provides a standardized set of deep mutational scanning datasets, enabling the most direct comparisons among these methods. Spearman correlation is the standard metric.

- **Fitness landscape estimation:** FLIP (Dallago et al., 2021) benchmarks (used by Cramming) evaluate on specific protein fitness tasks (GB1, AAV, Meltome) with Spearman correlation.

- **Structure prediction:** CASP (Kryshtafovych et al., 2021) and CAMEO (Haas et al., 2018) (used by PTQ4Protein) use TM-score and IDDT as metrics; these are not directly comparable to sequence-level metrics.

- **Functional annotation:** GO term prediction and enzyme commission number classification (used by ProtGO, MTDP) evaluate with $F_{\max}$ primarily, top-1 accuracy and weighted F1 score.

- **Protein-protein interaction:** PPI classification (used by Cramming) uses AUPRC.

- **Homology search:** PLMSearch and DHR are evaluated on sensitivity at fixed false-positive rates against alignment-based methods.

Because these benchmarks measure fundamentally different biological tasks, readers should exercise caution when comparing efficiency gains across methods evaluated on different benchmarks. We recommend that future efficiency-focused pLM research adopt standardized evaluation suites, such as ProteinGym for variant effect prediction and CASP/CAMEO for structure prediction, and report FLOPs and VRAM budgets alongside task-specific accuracy metrics, to enable more meaningful cross-method comparisons.

## 7 Future Directions

### 7.1 Near-Term Emerging LLM Techniques for pLMs

Since much of pLM development has drawn from the general language modeling literature, it is instructive to identify LLM techniques that have demonstrated empirical gains in NLP and are now beginning to be adapted for pLMs, though significant opportunities for deeper exploration remain:

- **Local and sparse self-attention:** Methods such as Longformer (Beltagy et al., 2020) and BigBird (Zaheer et al., 2020) combine local sliding-window attention with sparse global tokens to achieve linear-time complexity. While CARP and ProtHyena offer sub-quadratic alternatives through convolutions, the explicit local+global attention paradigm has seen initial adoption in pLMs, notably in Longformer-based protein encoders (Filipavicius et al., 2020) and biologically informed sparse attention mechanisms for viral-scale sequences (Dejean et al., 2025), yet remains underexplored relative to its potential for capturing both local secondary structure and long-range tertiary contacts.

- **Reinforcement learning from experimental feedback (RLXF):** Reinforcement learning from human feedback (Ouyang et al., 2022) has transformed LLM alignment, and an analogous paradigm is now emerging for pLMs. Recent work has replaced human preferences with experimental fitness measurements, stability assays, and binding affinity data as reward signals to fine-tune generative pLMs via PPO (Blalock et al., 2025) and GRPO (Wang et al., 2025b). Early results are promising, for example, substantial improvements in fluorescent protein brightness and binding affinity have been reported, but the design of effective reward models, the scalability of online RL loops for protein design, and comparisons with offline alternatives such as DPO remain active research questions.

- **Mixture of Experts (MoE):** Sparse MoE architectures (Fedus et al., 2022) enable conditional computation, activating only a subset of parameters per input. AIDO.Protein (Sun et al., 2024) is the first MoE model in the protein domain, featuring 16 billion parameters trained on over 1.2 trillion amino acids, with each token activating only 28% of total parameters via top-2 expert routing. It achieves state-of-the-art results across diverse benchmarks including ProteinGym DMS assays and structure-conditioned sequence generation. However, systematic studies of expert routing behavior, optimal expert granularity for biological data, and the extent to which individual experts specialize in distinct protein families or structural classes remain in early stages.

- **Efficient attention kernels:** Flash Attention (Dao et al., 2022) and related IO-aware implementations deliver substantial wall-clock speedups without approximation. Recent benchmarking efforts such as ESME (Çelik & Xie, 2025) and FAPLM (Fred Zhangzhi Peng, 2024) have demonstrated 4–9× faster inference and up to 14× memory reduction for ESM-2-family models, particularly when combined with sequence packing strategies that address the extreme length variability of protein batches. Broader adoption across pLM codebases and evaluation of newer variants (e.g., FlashAttention-3) remain ongoing.

- **Speculative decoding:** For autoregressive pLMs such as ProGen (Madani et al., 2023) and ProtGPT2 (Ferruz et al., 2022), speculative decoding (Leviathan et al., 2023) has recently been shown to accelerate sequence generation by 20–40% (Provatas et al., 2026) using lightweight draft models, with biologically informed drafting strategies such as k-mer-guided speculation (Walton et al., 2025) further improving acceptance rates. Scaling these approaches to high-throughput library generation and optimizing draft-to-target model ratios for protein-specific vocabularies are promising directions.

Although each of these techniques has now seen initial application in the protein domain, substantial opportunities remain. Protein sequences have distinct statistical properties, a small vocabulary, strong evolutionary correlations, and three-dimensional structural constraints, that necessitate domain-specific adaptations beyond straightforward transfer from NLP. We view the continued maturation and systematic benchmarking of these methods as a promising frontier for the pLM community.

### 7.2 Longer-Term Directions: Quantum Computing.

Beyond the classical efficiency techniques discussed above, quantum computing represents a fundamentally different computational paradigm that may offer advantages for specific sub-problems in protein modeling. While still in its early stages, we review this emerging direction to provide a forward-looking perspective.

**Quantum Algorithms for Protein Structure Prediction**  A central challenge in classical, physics-driven approaches to protein folding (like molecular dynamics (MD)) is their immense computational complexity. These methods simulate the intricate energy landscape of a protein, a task that scales poorly on classical hardware. This has motivated the exploration of quantum algorithms, which are naturally suited for simulating complex systems. An overview of quantum–classical learning pipeline is provided in Figure 9.

The core physical principle of protein folding is that a protein chain will configure itself into its most stable, lowest-energy state. In physics, the total energy of a system is described by a mathematical operator known as a *Hamiltonian*. Therefore, a protein's most stable 3D structure can be *reframed* as finding the *lowest possible energy value* of this Hamiltonian. This lowest-energy solution is known as the *ground state*.

Quantum Protein Structure Prediction (QPSP) attempts to solve this problem by first simplifying it (Robert et al., 2021; Chandarana et al., 2023; Doga et al., 2024; Li et al., 2025). The vast, continuous space of all possible protein folds is too complex to model directly. Instead, the protein is "discretized" by placing its residues onto the points of a 3D grid, or *lattice*. Common choices include cubic lattice (Wong & Chang, 2021; Babej et al., 2018; Fingerhuth et al., 2018) and tetrahedral lattice, which build upon earlier 2D models (Perdomo et al., 2008; Perdomo-Ortiz et al., 2012; Babbush et al., 2014).

With the problem now on a lattice, the Hamiltonian is assembled from terms that encode the structural energy: terms that capture the interactions between nearby residue and terms that act as constraints to enforce a structurally valid chain. Finding the ground state of this lattice-based Hamiltonian becomes an optimization task for a quantum computer. Research in this emerging field is focused on improving the realism of these models, progressing from 2D to 3D lattices, and developing more efficient encoding (e.g., turn-based rather than coordinate-based) to reduce the number of qubits required.

A fundamental limitation remains: native protein structures are not confined to a discrete lattice. This mismatch limits the fidelity of such models. Consequently, the accuracy of any lattice-based prediction is ultimately constrained by the resolution of the lattice, thereby necessitating models with higher degrees of freedom to yield more realistic fits (Godzik et al., 1993).

**Hybrid quantum–classical Transformers as a path toward quantum PLMs.**  A more immediate and pragmatic future direction lies in creating *hybrid quantum-classical models*. Instead of attempting to replace the entire classical pipeline, this approach seeks to identify computationally expensive sub-components of existing models (like PLMs) and "enhance" them with a quantum subroutine.

The self-attention mechanism is a prime target. At its core, the attention score is a similarity measure, typically a simple dot product, between a *Query (Q)* vector and a *Key (K)* vector. This score determines how much influence one token has on another. Recent work on hybrid transformers (Smaldone et al., 2025) proposes the replacement of this classical dot product.

In this hybrid model, the Q and K vectors are used to prepare quantum states. A quantum circuit then measures the *fidelity* (or *overlap*) between these two states, and this measurement becomes the new attention score. This is not just a faster way to compute a dot product; it is a *fundamentally different, quantum-native similarity metric.*

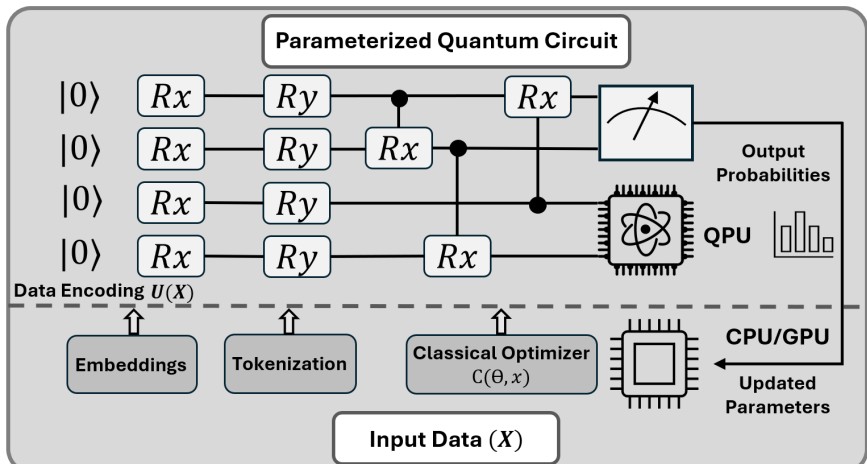

Figure 9: Conceptual illustration of a hybrid quantum–classical learning framework. Input data $X$ are first processed through classical preprocessing steps such as tokenization and embedding, and subsequently encoded into a parameterized quantum circuit via a data-dependent unitary $U(X)$. The quantum circuit consists of trainable single-qubit rotation gates and entangling operations, and is executed on a quantum processing unit (QPU), where measurements produce output probability distributions. These measurement outcomes are used to evaluate a classical objective function $C(\theta, x)$, whose feedback is computed on classical hardware (CPU/GPU) to iteratively update the circuit parameters $\theta$. This variational optimization loop illustrates a potential direction for integrating quantum computing with machine learning in future work.

This pathway is promising for several reasons. As *a new similarity metric*, this quantum-derived similarity is a specialized nonlinear function that may be able to capture complex relationships between token embeddings that a simple dot product misses, potentially improving model generalization in the low-data regimes common to protein engineering. It also offers *theoretical efficiency*: while the primary benefit may be representative power, this quantum subroutine also offers a theoretical cost reduction for the *scoring* portion, from $O(n^2 d)$ to $O(n^2 \log d)$, where $n$ is the sequence length and $d$ the embedding dimension. Furthermore, it has *NISQ-era (Noisy Intermediate-Scale Quantum) feasibility*, as the model remains "mostly classical," retaining standard tokenization, embedding, and feed-forward blocks. Crucially, the quantum subroutine only requires a qubit count that scales as $\log d$, making its parameter-efficiency a viable candidate for NISQ hardware. Finally, it represents *a practical prototyping path* that can be prototyped *today*. Using frameworks like CUDA-Q (NVIDIA Corporation, 2025b;a), the quantum circuit can be simulated on classical CPU/GPU hardware and integrated directly into standard ML training loops, allowing rapid, end-to-end validation of the concept.

Positive results from such simulations would position hybrid attention as a credible and practical first step toward future, quantum-enhanced PLMs for protein modeling.

## 8   Conclusion

In this survey, we systematically reviewed advancements in efficient pLMs across four dimensions: dataset optimization, model architecture, training strategies, and inference. We highlighted a spectrum of techniques, from meta-learning and scaling law-guided design to PEFT, knowledge distillation, and quantization, that collectively address the formidable computational demands of modern pLMs. These approaches enable substantial cost reductions, democratize accessibility, and lay the groundwork for scalable, practical deployment in computational biology.

Despite this substantial progress, significant challenges remain. While efficient pLMs achieve competitive results on standard benchmarks, a performance gap often emerges in complex tasks such as de novo structure prediction or out-of-distribution generalization. The optimal strategies for downstream adaptation, such as fine-tuning protocols and representation pooling in extreme resource-limited settings, also remain underex-

plored. Future research must not only extend these efficiency paradigms to more diverse biological tasks but also systematically benchmark adaptation strategies and develop even more scalable solutions through advances in optimization, quantization, and architectural innovation.

We anticipate that continued innovation in these areas, coupled with broader community engagement, will be essential for unlocking the full potential of efficient pLMs and accelerating biological discovery. Our cross-cutting analysis reveals that the most effective efficiency gains emerge from combining techniques across dimensions (e.g., distillation with quantization, or meta-learning with parameter-efficient fine-tuning). We have also identified several promising LLM techniques, including sparse attention, mixture of experts, RLHF-style optimization, and speculative decoding, that remain largely unexplored for pLMs and represent concrete opportunities for future work. As pLMs are increasingly deployed as components in iterative agentic workflows for protein design and discovery, the practical importance of inference efficiency and model compactness will only grow.

## Glossary

**3Di** Three-Dimensional Interaction (token). 17

**AA** Amino Acid. 1, 7, 12, 17

**Cryo-EM** Cryo-Electron Microscopy. 1

**cubic lattice** A simple three-dimensional lattice in which residues occupy vertices of a cube and can move to six nearest neighbors, used as a basic discretization model in coarse-grained protein folding. 27, *see also* lattice

**energy landscape** A multidimensional surface representing the potential energy of a molecular system as a function of its atomic coordinates; valleys correspond to stable conformations and barriers to transition states. 27

**Enzyme Commission (EC) number** A numerical classification code for enzymes, based on the chemical reactions they catalyze.. 21

**functional genomics** A field of molecular biology that aims to understand the function of genes and proteins, often at a genome-wide scale.. 8

**GFP** Green Fluorescent Protein. 19

**GO** Gene Ontology. 21

**Hamiltonian** The energy operator of a system, representing the total energy (kinetic plus potential) and governing its evolution in classical or quantum mechanics; used to encode energy terms in protein folding and quantum models. 27

**homolog detection** The computational task of identifying proteins that share a common evolutionary ancestor (homologs), which is often challenging when sequence similarity is low (remote homologs).. 22, 24

**homology search** A bioinformatics method used to identify sequences with a shared evolutionary ancestry, typically by comparing a query sequence against a database.. 23, 24

**lattice** A discrete grid (2D or 3D) used in coarse-grained protein folding models where residues occupy vertices and moves are restricted to adjacent lattice points. Common choices include square, cubic, and face-centered cubic (FCC) lattices, which approximate steric constraints and allow efficient enumeration of conformations. 27

**metagenomic** Relating to the study of genetic material recovered directly from environmental samples, which often contain a vast collection of uncultured and unknown organisms.. 7, 9

**molecular dynamics (MD)** A computational simulation method that models the physical movements of atoms and molecules over time by numerically integrating Newton's equations of motion, often used to explore protein folding and conformational changes. 27

**MSA** Multiple Sequence Alignment. 9, 19, 21, 23, 24

**NISQ** An acronym for *Noisy Intermediate-Scale Quantum*, referring to the current generation of quantum devices that contain tens to a few hundred qubits. These systems are powerful enough to perform small-scale quantum computations but still suffer from noise, decoherence, and limited qubit connectivity.. 28

**NMR** Nuclear Magnetic Resonance Spectroscopy. 1

**proteome** The complete set of proteins expressed by an organism, cell, or tissue at a given time.. 17, 23

**qubit** The fundamental unit of quantum information, represented by a two-level quantum system that can exist in superposition of basis states $|0\rangle$ and $|1\rangle$; used to encode residue interactions or Hamiltonian terms in quantum protein models. 28

**remote homology detection** The task of identifying distant evolutionary relationships between proteins, typically when their sequence similarity is very low but their 3D structures are still related.. 17

**residue** An individual amino acid unit within a protein chain; in lattice and coarse-grained models, each residue is mapped to a vertex or bead. 27

**secondary structure prediction** The task of predicting the local secondary structures of a protein (e.g., alpha-helices, beta-sheets) from its primary amino acid sequence.. 17, 19

**structural energy** The potential energy associated with a given molecular or protein conformation, determined by intra- and inter-residue interactions, often minimized during structure prediction or folding simulations. 27

**subcellular localization** The specific location or compartment within a cell where a protein resides and performs its function (e.g., nucleus, mitochondria, cytoplasm).. 22

**tetrahedral lattice** A three-dimensional lattice where each vertex connects to four nearest neighbors forming tetrahedral geometry, offering improved angular representation of protein backbones compared to cubic lattices. 27, *see also* lattice

**variant effect prediction** The task of predicting the functional impact (e.g., on fitness, stability) of a mutation or a set of mutations on a protein.. 21

**X-ray crystallography** A foundational experimental technique used to determine the precise three-dimensional atomic structure of a protein by analyzing the diffraction pattern of X-rays passing through a protein crystal.. 1

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
