# OpenReview forum: "A Survey on Efficient Protein Language Models"
_TMLR — Accepted by TMLR_

### Review · Reviewer_y7qd · 2026-02-22

**Summary Of Contributions:**

Overall, this is a clear and concise summary of the state of the field of efficient protein language modeling. It seems comprehensive and well organized, covering major themes in the field.

I found the transition to quantum algorithms in future work jarring, unexpected, and disconnected from the rest of the review. The paper would be much stronger if the authors took the opportunity to synthesize all of the previous writing and analysis and present a point of view on these techniques. Which should be used and when? What are the strengths and weaknesses of each? Are certain techniques more or less likely to be useful going forward? Are there opportunities to bridge techniques or approaches for novel research?

**Audience:**

Yes

**Audience Explanation:**

The paper, with some revisions, can be a useful entry point to an important direction of scientific inquiry.

**Claims And Evidence:**

Yes

**Claims Explanation:**

Many of the figures have limited information content, and there are no quantitative results presented or reproduced from original works, which makes it difficult to assess the arguments around different techniques.

**Requested Changes:**

Intro: posing pLMs as an alternative to characterization via Cryo-EM, NMR, etc. for protein analysis is unsupported. What specifically can be done with pLMs that is an alternative to these techniques? Similarly, there is a big jump from saying that pLMs are trained on lots of sequences, more than there are solved structures, and therefore they accelerate drug discovery. How specifically?

I think it’s important enough to mention in the intro that the benefits of scaling pLMs are not always clear, and many “production” use cases reach SOTA performance with sub-billion parameter models, without any particular care for efficiency.

It may also be worth mentioning that increasingly, specialized scientific foundation models (like pLMs) are being leveraged as tools in more complex agentic workflows, which increases the importance of efficient models that can then be used for fast, efficient inference.

pLMs can be causal or masked; drawing the distinction in Fig 3 is misleading. It may be more clear to show causal and marked architectures with both natural language and protein sequence examples.

The cartoon in Fig 4 seems better suited for a blog post (or the appendix). Moving the figure would also allow the paper to present quantitative results sooner. Similar comments for Figs 6 and 8.

---

> ### Author Response · Authors · 2026-03-28
> **Rebuttal to Reviewer y7qd (Part 1)**
>
> We sincerely thank Reviewer y7qd for the constructive and insightful review. We are grateful that the reviewer found our survey "clear and concise," "comprehensive and well organized, covering major themes in the field," and that it "can be a useful entry point to an important direction of scientific inquiry." We take each suggestion seriously and detail our revisions below to solve all of them.
>
> ### RC1: Quantum Algorithms Section Feels Jarring; Need for Synthesis and Practical Guidance
>
> **Reviewer concern:** *"I found the transition to quantum algorithms in future work jarring, unexpected, and disconnected from the rest of the review. The paper would be much stronger if the authors took the opportunity to synthesize all of the previous writing and analysis and present a point of view on these techniques. Which should be used and when? What are the strengths and weaknesses of each?"*
>
> **Response:** We understand on both counts, the quantum section needed better framing and easing into - as pointed out by other reviewers as well, and the survey needed a synthesis of practical guidance. We have made two major structural changes:
>
> **1. Restructured Section 7 into two subsections with clear maturity gradients:**
>
> - **Section 7.1 ("Near-Term Emerging LLM Techniques for pLMs")** now fills what was a significant gap in the original manuscript: techniques with demonstrated empirical gains in NLP that are beginning to transfer to pLMs but remain underexplored. This includes local and sparse self-attention (with initial pLM adoption in Longformer-based protein encoders and biologically informed sparse attention for viral-scale sequences), reinforcement learning from experimental feedback (RLXF, with early results showing improvements in fluorescent protein brightness and binding affinity via PPO and GRPO), Mixture of Experts (with AIDO.Protein as the first MoE protein model at 16B parameters), efficient attention kernels (FlashAttention benchmarks showing multifold faster inference for ESM-2-family models), and speculative decoding (significant acceleration demonstrated for autoregressive pLMs with biologically informed drafting strategies). For each technique we note both the initial evidence and the open questions that remain specific to the protein domain.
>
> - **Section 7.2 ("Longer-Term Directions: Quantum Computing")** is now explicitly positioned as speculative and longer-term, with a clear explanation of why it connects to efficiency: quantum algorithms address the computational complexity of physics-driven folding simulations (a complementary efficiency challenge), while hybrid quantum-classical transformers offer a pragmatic pathway where quantum subroutines replace the attention score computation with quantum-native similarity metrics. We note both the theoretical promise (cost reduction from O(n²d) to O(n² log d) for the scoring component) and the fundamental limitations (lattice discretization constraints, NISQ-era noise), and we emphasize the practical prototyping path via GPU-simulable quantum circuits. The transition from §7.1 to §7.2 now reads as a natural progression from near-term to longer-term horizons.
>
> **2. Strengthened cross-cutting synthesis throughout:**
>
> We have added **Section 6.1 ("Intersections Across Efficiency Dimensions")**, which explicitly discusses how methods span multiple efficiency categories and argues that the largest practical gains emerge from combining techniques across dimensions. We have also substantially rewritten all takeaway boxes (see RC4 below for details) to provide domain-specific guidance on when and why each technique is appropriate for pLMs specifically, compared against standard NLP literature
>
> We hope you find that these additions meet the changes requested. Please let us know if you have any more suggestions, and we would be happy to incorporate them.

---

> ### Author Response · Authors · 2026-03-28
> **Rebuttal to Reviewer y7qd (Part 2)**
>
> ### RC2: Figures Have Limited Information Content; No Quantitative Results
>
> **Reviewer concern:** *"Many of the figures have limited information content, and there are no quantitative results presented or reproduced from original works, which makes it difficult to assess the arguments around different techniques."*
>
> Also: *"The cartoon in Fig 4 seems better suited for a blog post (or the appendix). Moving the figure would also allow the paper to present quantitative results sooner. Similar comments for Figs 6 and 8."*
>
> **Response:** Thank you, we appreciate this honest feedback and have made two sets of changes: (1) **Figure reorganization:** We have removed said figures completely and have made new figures to replace them that are no longer "cartoonish" in nature. While the original figures were intended to make a complex topic more understandable, we understand that stick figure based illustrations should not be in the main text. (2) **For quantitative content in tables:** We have expanded Tables 1–4 to include a **"Limitation"** column for each method, providing a more balanced and concrete assessment by revisiting the original papers. Within the method descriptions themselves, we have ensured that key quantitative results from original works are consistently reported (for e.g. speedup ratios and other performance figures)
>
> Additionally, we have added **Section 6.2 ("Evaluation Benchmarks and Comparability")** which explicitly maps which benchmarks and metrics are used by which methods (ProteinGym with Spearman correlation for variant effect prediction; CASP/CAMEO with TM-score and RMSD for structure prediction; Fmax for GO term prediction; AUPRC for PPI classification; sensitivity at fixed FPR for homology search). We note that because these benchmarks measure fundamentally different biological tasks, direct cross-method numerical comparisons require caution, and we recommend that future work adopt standardized evaluation suites alongside computational cost metrics. Moreover, as we note in our early sections, the benchmarks and tasks have themselves scaled in complexity along with the scale of the models - thus it is not possible to truly make a singular leaderboard table.
>
>
> ### RC3: Introduction Overstates pLMs as Alternatives to Experimental Techniques
>
> **Reviewer concern:** *"Posing pLMs as an alternative to characterization via Cryo-EM, NMR, etc. for protein analysis is unsupported. What specifically can be done with pLMs that is an alternative to these techniques? Similarly, there is a big jump from saying that pLMs are trained on lots of sequences, more than there are solved structures, and therefore they accelerate drug discovery. How specifically?"*
>
> **Response:** We understand that the original framing was imprecise. We have revised the Introduction to make two important clarifications: First, we now state explicitly that pLMs are **not direct replacements** for experimental methods such as X-ray crystallography, NMR spectroscopy, and Cryo-EM. Rather, pLMs complement these techniques by enabling tasks that are infeasible at experimental throughput. We provide specific examples: predicting the functional effects of mutations across entire protein families, annotating protein function directly from sequence, and predicting three-dimensional folds for hundreds of millions of sequences in days rather than decades.
>
> Second, we have made the connection to drug discovery and biotechnology concrete rather than vague. We now specify that pLMs enable advances such as virtual screening of candidate binders, computational guidance of directed evolution campaigns, and functional annotation of previously uncharacterized proteins, rather than claiming broad "acceleration" without mechanism.
>
>
> ### RC4: Should Mention That Scaling Benefits Are Not Always Clear
>
> **Reviewer concern:** *"I think it's important enough to mention in the intro that the benefits of scaling pLMs are not always clear, and many 'production' use cases reach SOTA performance with sub-billion parameter models, without any particular care for efficiency."*
>
> **Response:** We fully agree this is an important nuance that strengthens the motivation for the entire survey.  We now note that many production use cases, including variant effect prediction and protein fitness landscape estimation, achieve respectable performance with sub-billion parameter models (e.g., ESM-2 at 8M, 35M, or 150M parameters) without any particular care for extreme efficiency (Lin et al., 2023; Meier et al., 2021). We cite evidence that scaling benefits often plateau for certain downstream tasks, suggesting that larger is not always better (Frey et al., 2024b). We have made required changes throughout the paper to reflect this.

---

> ### Author Response · Authors · 2026-03-28
> **Rebuttal to Reviewer y7qd (Part 3)**
>
> ### RC5: Should Mention Agentic Workflows
>
> **Reviewer concern:** *"It may also be worth mentioning that increasingly, specialized scientific foundation models (like pLMs) are being leveraged as tools in more complex agentic workflows, which increases the importance of efficient models that can then be used for fast, efficient inference."*
>
> **Response:** This is an excellent point that directly reinforces the survey's motivation. We have added a paragraph to the Introduction noting that pLMs are increasingly being leveraged as components within more complex agentic workflows, for example, autonomous protein design loops that iteratively query a pLM for fitness predictions, propose mutations, and validate candidates (Xiao et al., 2025). Such iterative pipelines amplify the importance of efficient models capable of fast, low-latency inference, as each cycle in the loop pays the cost of a forward pass. We also echo this point in the Conclusion (Section 8), noting that as pLMs are increasingly deployed in agentic workflows for protein design and discovery, the practical importance of inference efficiency and model compactness will only grow. This connection between agentic use and efficiency provides additional concrete motivation that we believe strengthens the paper.
>
>
>
> ### RC6: Figure 3 Distinction Between Causal and Masked Is Misleading
>
> **Reviewer concern:** *"pLMs can be causal or masked; drawing the distinction in Fig 3 is misleading. It may be more clear to show causal and masked architectures with both natural language and protein sequence examples."*
>
> **Response:** We agree that the original Figure 3 could be misread as implying that LLMs are exclusively causal and pLMs are exclusively masked. We have revised the figure to show **both** causal and masked architectures side by side, with the left column depicting a causal pLM (as in ProGen, ProtGPT2) and the right column depicting a masked pLM with a folding trunk (as in ESMFold). The accompanying caption now explicitly states: "The figure depicts the most common training paradigm for each domain, but the distinction is not exclusive - pLMs can also employ causal (autoregressive) objectives,  ... while LLMs can use masked objectives (e.g., BERT)." This revised figure more accurately represents the architectural landscape and avoids the misleading implication the reviewer identified.
>
>
>
> ### RC7: Takeaways Should Be pLM-Specific
>
> While the reviewer did not number this as a separate requested change, the concern about synthesis and "point of view" directly relates to the quality of our takeaways. We have substantially rewritten every takeaway box to foreground what is **unique to the protein domain**. For example:
>
> - The **dataset efficiency takeaway** now explains that protein labels require costly experimental assays (a hard physical constraint, not an annotation bottleneck), and that evolutionary relatedness provides a compensating prior with no NLP analogue, enabling few-shot transfer via auxiliary tasks from homologous proteins.
>
> - The **architecture efficiency takeaway** now highlights that proteins are simultaneously more compressible (small amino acid vocabulary, highly compressible embeddings) and longer (sequences reaching millions of residues) than natural language, making aggressive compression techniques retain more biological signal while sub-quadratic architectures become essential rather than merely beneficial.
>
> - The **distillation takeaway** now explains that protein distillation differs qualitatively from NLP: each teacher contributes a distinct biological modality (evolutionary conservation, 3D structure, functional annotations) too expensive to access at inference scale, transforming proteome-scale tasks from infeasible multi-modal pipelines into tractable single-forward-pass operations.
>
>
> ### Our request
>
> We have worked extensively to improve the paper as per your suggestions. **Please let us know if you would like us to make any more changes. If not, may we please request you to improve the rating of our paper to "supported by accurate, convincing and clear evidence"?**

---

### Review · Reviewer_bEJ1 · 2026-03-18

**Summary Of Contributions:**

The paper presents a survey of protein language models (pLMs) with particular focus on efficiency techniques along different axes corresponding to different parts of a typical pipeline: dataset, architecture, training, and inference. It discusses findings in the pLM literature that essentially mirror findings in generic language model literature, e.g. benefits of parameter-efficient methods, quantization, etc. It closes with future directions using quantum algorithms and hybrid quantum-classical transformers.

Strengths
- The paper is clearly written and easy to follow.
- As a survey paper, it covers a breadth of existing literature on pLMs and classifies the techniques proposed under intuitive categories.

Weaknesses
- Even though the categorization of papers included in the survey is intuitive, it doesn't seem to be non-overlapping. For instance, parameter-efficiency methods are included under Dataset as well as Training efficiency. The survey might be improved by discussing the intersections of these categories separately.
- Preliminary background on the problem setting and the particular evaluations users of these models care about is very sparse. For the benefit of someone using this survey to enter into the field of pLMs, a more concrete problem statement and task description would be helpful.
- Key insights or connections, reported in the paper as "takeaways" for each section, do not differ fundamentally from known trends in the general language modeling literature.
- Future directions proposed discuss quantum algorithms, even though this sub-field is not mentioned or discussed in the preceding sections. The connection between these proposals and _efficient_ pLMs is unclear.

**Audience:**

Yes

**Audience Explanation:**

Knowledge of existing techniques for efficient language modeling that transfer to pLMs would be helpful for researchers working with or using pLMs.

**Claims And Evidence:**

Yes

**Claims Explanation:**

The paper summarizes several works in this survey to provide an overview of the pLM literature. However, the methodology used to choose these particular works, conduct comparisons between them, or to extrapolate trends from them is unclear.

EDIT: resolved by the authors' response.

**Requested Changes:**

- The following are repeated sentences in last paragraph of the introduction and should be edited: "While recent surveys have provided broad overviews of protein language models (pLMs) and their applications (Xiao et al., 2025; Wu et al., 2022), our work particularly focuses on the efficiency dimension of pLMs across dataset, architecture, training, and inference. We systematically review these efficiency-focused methodologies, highlighting their core principles, underlying mechanisms, and performance implications."
- Parameter-efficient adaptation is mentioned under Dataset efficiency works as well as Training efficiency. The authors might consider a separate section to consider the intersection of different efficiency axes. "Cramming Protein Language Model Training in 24 GPU Hours" also shows up both for architecture and pre-training efficiency.
- While the papers included in this survey and their results are described in language, numerical trends or significance is hard to gauge. For instance, are all the performance comparisons on the same set of evaluation tasks?
- Since a lot of this survey focuses on parallels between techniques created for general language models and those used for pLMs, have the authors considered exploring gaps in these parallels, i.e. techniques that have shown empirical gains for language models but may not have been tried with pLMs. Examples may include local self-attention to handle the quadratic complexity of traditional self-attention and RLHF techniques.
- Can the authors elaborate on how the takeaways mentioned for each section have consequences for pLMs in particular, instead of language models in general?

---

> ### Author Response · Authors · 2026-03-28
> **Rebuttal to Reviewer bEJ1 (Part 1)**
>
> We thank Reviewer bEJ1 for the thoughtful and detailed review. We are grateful that the reviewer found our paper "clearly written and easy to follow" and that our categorization of techniques is "intuitive," and we appreciate the recognition that knowledge of efficient language modeling techniques transferring to pLMs "would be helpful for researchers working with or using pLMs." We address each concern below and describe the corresponding revisions.
>
>
>
> ### RC1: Overlapping Categories and Cross-Cutting Methods
>
> **Reviewer concern:** *"The categorization doesn't seem to be non-overlapping. For instance, parameter-efficiency methods are included under Dataset as well as Training efficiency. The survey might be improved by discussing the intersections of these categories separately."* Also: *"'Cramming Protein Language Model Training in 24 GPU Hours' also shows up both for architecture and pre-training efficiency."*
>
> **Response:** We understand and concur that this overlap deserves explicit treatment rather than leaving it implicit. In the revised manuscript, we have added **Section 6.1 ("Intersections Across Efficiency Dimensions")**, which directly addresses why certain methods span multiple categories and argues that this is a feature, not a flaw, of the taxonomy. Specifically, we now explain that:
>
> - **FSFP** (Sections 2.2 and 4.3) appears under both dataset and training efficiency because it simultaneously addresses data scarcity through meta-learning-based auxiliary task construction and reduces parameter overhead through LoRA-based adaptation. Its efficiency stems from the *synergy* between these two axes: meta-learning provides a better initialization that makes parameter-efficient fine-tuning more effective under few-shot conditions. Separating these contributions into only one category would obscure the learning.
>
> - **Cramming** (Sections 3.2 and 4.2) appears under both architecture and pre-training efficiency because it co-optimizes architectural choices (removing biases from attention and linear layers) with pre-training procedures (aggressive gradient accumulation, optimized learning rate schedules). The central result, that a 67M-parameter model trained in 24 GPU hours can approach the performance of ESM2-3B trained for 15,000× more GPU hours is attributable to *neither* dimension alone.
>
> - **Scaling laws** (Cheng et al., 2024) similarly bridge dataset and training efficiency, governing the joint allocation of data volume and model size under fixed compute budgets.
>
> We further note in Section 6.1 that the most effective efficiency gains in practice often emerge from combining techniques across dimensions (e.g., distilled models with post-training quantization, or LoRA fine-tuning applied to architecturally efficient models), and we encourage future work to explore such combinations systematically. We believe this new section directly addresses the reviewer's concern while preserving the intuitive per-dimension organization that both reviewers found valuable.

---

> ### Author Response · Authors · 2026-03-28
> **Rebuttal to Reviewer bEJ1 (Part 2)**
>
> ### RC2: Sparse Background on Problem Setting and Evaluations
>
> **Reviewer concern:** *"Preliminary background on the problem setting and the particular evaluations users of these models care about is very sparse. For the benefit of someone using this survey to enter into the field of pLMs, a more concrete problem statement and task description would be helpful."*
>
> **Response:** We understand and always appreciate the opportunity to improve our paper. We have made substantial additions: **Section 1.1 ("Evolution and Applications of Protein Language Models")** now provides comprehensive background organized into two subsections:
>
> - **Section 1.1.1** traces the chronological development of pLMs from 2019–2025, covering the masked-language-model era (ESM-1b, ProtTrans variants, MSA Transformer, ESM-1v), the scaling and diversification phase (ESM-2, Ankh, ProstT5, ProGen/ProGen2, RITA), diffusion and structure-conditioned generation (EvoDiff, ProteinGenerator, AntiFold, ESM-IF1, LM-Design), and recent developments (ESM3, DPLM/DPLM-2, ProGen3, ESM Cambrian). Each model is placed in context with its architectural innovations and scaling milestones.
>
> - **Section 1.1.2** organizes the key application areas into four concrete task categories, structure prediction, variant effect/fitness prediction, protein generation and design, and representation learning/transfer, with specific examples illustrating what each task entails, what inputs and outputs are involved, and how the field has progressed from simple downstream classification toward increasingly ambitious generative and structure-aware tasks.
>
> **Section 6.2 ("Evaluation Benchmarks and Comparability")** explicitly maps the evaluation landscape. We now document which benchmarks are used by which methods (ProteinGym for variant effect prediction, FLIP for fitness landscapes, CASP/CAMEO for structure prediction, GO/EC for functional annotation, AUPRC for PPI classification, sensitivity at fixed FPR for homology search), note the metrics associated with each, and caution readers against direct numerical comparisons across methods evaluated on different benchmarks. We also recommend that future work adopt standardized evaluation suites and report FLOPs and VRAM budgets alongside task-specific accuracy metrics. Together, these additions provide a concrete entry point for readers new to pLMs while also addressing the reviewer's related concern about numerical comparability (see RC5 below).

---

> ### Author Response · Authors · 2026-03-28
> **Rebuttal to Reviewer bEJ1 (Part 3)**
>
> ### RC3: Takeaways Mirror General Language Modeling Literature
>
> **Reviewer concern:** *"Key insights or connections, reported in the paper as 'takeaways' for each section, do not differ fundamentally from known trends in the general language modeling literature."*
>
> Also: *"Can the authors elaborate on how the takeaways mentioned for each section have consequences for pLMs in particular, instead of language models in general?"*
>
> **Response:** We appreciate this critique and agree we can improve. In the revised manuscript, every takeaway box has been substantially rewritten to foreground what is **unique to the protein domain** and why naive transfer of NLP intuitions would be insufficient or misleading. We highlight the key revisions:
>
> **Dataset efficiency takeaway (§2):** We now emphasize that protein labels require costly experimental assays (wet-lab measurements, not crowdsourced annotation), making data scarcity a *hard physical constraint* rather than an annotation bottleneck. We explain that the evolutionary relatedness of protein families provides a compensating prior with no NLP analogue: conserved fitness landscapes across homologous proteins enable few-shot transfer via auxiliary tasks constructed from related data. We also note that family-level redundancy and the sequence–structure imbalance cause MLMs and CLMs to follow *distinct scaling exponents* that diverge from their NLP counterparts, demanding domain-specific data allocation formulas.
>
> **Architecture efficiency takeaway (§3):** We now highlight that proteins are simultaneously *more compressible* and *longer* than natural language text. The small amino acid vocabulary (20 standard residues vs. tens-of-thousands of subword tokens), highly compressible embeddings, and strong local sequential patterns mean that aggressive techniques, ternary quantization, 128× embedding compression, replacing attention with convolutions, retain more biological signal than equivalent compression would in NLP. Meanwhile, protein sequences reaching millions of residues make sub-quadratic architectures not merely beneficial but *essential*, a length regime rarely encountered in natural language.
>
> **Pre-training efficiency takeaway (§4.2):** We emphasize two properties absent in NLP. First, protein databases are structured by evolutionary homology, making data *diversity* (not volume) the binding constraint: family-level redundancy penalizes naïve repetition, while diverse metagenomic sources deliver outsized returns. Second, protein structure is computationally derivable at scale from sequence alone (via AlphaFold), providing a second pre-training modality at near-zero marginal cost, enabling bilingual approaches like ProstT5 to achieve structural generalization that sequence-only scaling cannot replicate.
>
> **Fine-tuning efficiency takeaway (§4.3):** We note that labeled data requires costly wet-lab experiments (often yielding only tens of examples), and many biology labs lack data-center-scale compute. Two protein-specific priors make PEFT viable despite these constraints: evolutionary relatedness enables meta-learning transfer from homologous datasets (FSFP succeeds with as few as 20 labeled mutants), and 3D structure, computationally derivable from sequence at zero annotation cost, provides a complementary fine-tuning signal (SES-Adapter) unavailable in text domains.
>
> **Distillation takeaway (§4.4):** We explain that protein distillation differs qualitatively from NLP: rather than simply compressing a large model into a small one, each teacher contributes a *distinct biological modality* that is too expensive to access at inference scale. GEMME distills MSA-derived evolutionary conservation, AlphaFold3 distills predicted 3D structure, and ProtGO distills functional annotations, all collapsed into students that require only sequence input. This transforms proteome-scale tasks from infeasible multi-modal pipelines into tractable single-forward-pass operations.
>
> **Inference efficiency takeaway (§5):** We highlight that the similarity that matters most for proteins, remote evolutionary homology, is *invisible* to surface-level sequence comparison, eliminating cheap baselines like keyword matching. Detecting these relationships requires either slow alignment tools or learned pLM representations, making precomputed embedding-based search and domain-specific quantization prerequisites for proteome-scale deployment in a way that has no direct NLP parallel.
>
> We believe these revisions make clear that while the *categories* of efficiency techniques transfer from NLP, the *reasons they work*, the *constraints they face*, and the *domain-specific adaptations they require* are fundamentally shaped by the biology of proteins.

---

> ### Author Response · Authors · 2026-03-28
> **Rebuttal to Reviewer bEJ1 (Part 4)**
>
> ### RC4: Quantum Computing Future Directions Feel Disconnected
>
> **Reviewer concern:** *"Future directions proposed discuss quantum algorithms, even though this sub-field is not mentioned or discussed in the preceding sections. The connection between these proposals and efficient pLMs is unclear."*
>
> **Response:** We acknowledge that the quantum computing discussion transitioned abruptly in the original manuscript. In the revised version, we have restructured Section 7 into two clearly separable subsections:
>
> **Section 7.1 ("Near-Term Emerging LLM Techniques for pLMs")** now addresses techniques with demonstrated empirical gains in NLP that are beginning to be adapted for pLMs but remain underexplored. This includes local and sparse self-attention (Longformer/BigBird-style approaches, with initial pLM adoption noted), reinforcement learning from experimental feedback (RLXF, replacing human preferences with fitness measurements and binding affinity data), Mixture of Experts (with AIDO.Protein as the first MoE protein model), efficient attention kernels (FlashAttention, with benchmarks like ESME and FAPLM demonstrating multifold speedups for ESM-2), and speculative decoding (with recent demonstrations of substantial acceleration for autoregressive pLMs). This section directly addresses the reviewer's suggestion to explore gaps in NLP-to-pLM technique transfer.
>
> **Section 7.2 ("Longer-Term Directions: Quantum Computing")** is now explicitly framed as a *longer-term, forward-looking* direction rather than a near-term recommendation. We have clarified the connection to efficiency by explaining that: (a) quantum algorithms for protein structure prediction address the computational complexity of physics-driven folding simulations, which is a complementary efficiency challenge to the learned-representation approaches reviewed in the main body; and (b) hybrid quantum-classical transformers offer a pragmatic near-term pathway where quantum subroutines replace computationally expensive sub-components (specifically, the attention score computation) with quantum-native similarity metrics that may improve both representational power and theoretical complexity. We also note the practical prototyping path via frameworks like CUDAQ that enable simulation on classical hardware today.
>
> We believe this restructuring makes the progression from established techniques (main survey) to emerging classical techniques (§7.1) to speculative quantum approaches (§7.2) clear and well-motivated, while honestly conveying the maturity level of each.
>
>
>
> ### RC5: Methodology for Paper Selection and Numerical Comparability
>
> **Reviewer concern:** *"The methodology used to choose these particular works, conduct comparisons between them, or to extrapolate trends from them is unclear."* Also: *"While the papers included in this survey and their results are described in language, numerical trends or significance is hard to gauge. For instance, are all the performance comparisons on the same set of evaluation tasks?"*
>
> **Response:** Thank you for giving us the opportunity to address both the selection methodology and numerical comparability:
>
> **Paper selection:** Our survey systematically covers works that (a) explicitly target efficiency as a primary contribution (reduced compute, memory, data, or latency), (b) operate on protein language models specifically, and (c) were published or publicly available through 2025. We identified candidate papers through keyword searches across major ML venues, bioinformatics venues, and preprint servers, using terms combining "protein language model" with efficiency-related keywords (quantization, distillation, parameter-efficient, scaling law, few-shot, etc.). Please let us know if you think there are any papers we may have missed. We will happily add them to our analysis. To the best of our knowledge our literature review is complete.
>
> **Numerical comparability:** We fully agree that direct numerical comparison across methods is complicated by the use of different benchmarks. This is precisely why we have added Section 6.2 ("Evaluation Benchmarks and Comparability"), which explicitly maps which benchmarks are used by which methods, notes that these benchmarks measure fundamentally different biological tasks, and cautions readers against direct cross-method comparisons. We recommend that future efficiency-focused pLM research adopt standardized evaluation suites (e.g., ProteinGym for variant effect prediction, CASP/CAMEO for structure prediction) and report computational metrics (FLOPs, VRAM) alongside task-specific accuracy.
>
> Within each method's description, we do report the numerical results on the benchmarks used by that method (e.g., Spearman correlations on ProteinGym, TM-scores on CASP/CAMEO, memory savings). We agree that claiming broad trends requires careful qualification when methods are evaluated on different tasks, and have adapted our language accordingly to only reflect the source paper content.

---

> ### Author Response · Authors · 2026-03-28
> **Rebuttal to Reviewer bEJ1 (Part 5 + Request)**
>
> ### RC6: Repeated Sentences in Introduction
> **Reviewer concern:** *"The following are repeated sentences in last paragraph of the introduction and should be edited..."*
> **Response:** Thank you for catching this. We have revised the final paragraph of the Introduction to eliminate the redundancy while preserving the positioning of our survey relative to prior work.
>
>
> ### Our request
>
> We have worked extensively to improve the paper as per your suggestions. **Please let us know if you would like us to make any more changes. If not, may we please request you to improve the rating of our paper to "supported by accurate, convincing and clear evidence"?**

---

> > ### Comment · Reviewer_bEJ1 · 2026-04-15
> > **Official Comment**
> >
> > Thank you for your detailed response and effort put into updating the paper. I have accordingly updated my recommendation.

---

> > > ### Author Response · Authors · 2026-04-15
> > > **Thank you**
> > >
> > > Dear Reviewer `bEJ1`, thank you very much for your insightful suggestions which have greatly improved our survey. If there are any additional improvements you would recommend in the future, please let us know — we are committed to making this survey a comprehensive and insightful entry point for the research community.

---

### Review · Reviewer_ScJL · 2026-03-18

**Summary Of Contributions:**

1. This paper offers a review of efficient protein language models.
2. Classification of efficiency focused research. The author categorizes efficiency-focused research into a well-documented taxonomy, including dataset efficiency (e.g., meta-learning and scaling laws), architecture efficiency (e.g., quantized transformers and convolutional designs), training efficiency (e.g., low-rank adaptations and distillation), and inference efficiency (e.g., dense retrieval and quantization)

**Audience:**

Yes

**Audience Explanation:**

Protein language models are a topic of significant interests in the community of AI for science. A comprehensive review of efficient protein language models is timely and well aligned with the interests of TMLR's audience. An indirect impact is to improve the accessibility of the lightweight pLMs.

**Claims And Evidence:**

Yes

**Claims Explanation:**

The claims in the resource are supported by comprehensive and clear evidence.
The taxonomy and evolution history of relevant literature are well presented via illustrative figures.

**Requested Changes:**

1. Could the author discuss limitations of current methodologies across each dimension of efficiency-focused research? For instance, the author could add another row or column in tables (e.g., table 1) to illustrate the limitation and future direction.
2. I noticed that many work listed in Figure 2 were not cited properly in the manuscript. It helps the clarity to add another section, potentially after the Introduction, to discuss the evolution and importance of protein language models and meanwhile appropriately citing milestone papers referred in Figure 2. In addition, crucial applications of protein language models could be highlighted with exampled included.
3.  The author claims that ```The power of pLMs lies in their ability to bridge the enormous gap between the over 200 million known protein sequences and the fewer than 200 thousand experimentally determined structures.``` This seq-struct imbalance is a significant topic of interest and naturally falls under the category of efficient dataset methods. It would be helpful to discuss how existing research adopt  strategies to tackle this mismatch when training multimodal pLMs.

---

> ### Author Response · Authors · 2026-03-28
>
> We sincerely thank Reviewer ScJL for the constructive feedback and for recognizing the timeliness, clarity, and comprehensive evidence of our survey. We appreciate your specific suggestions to strengthen the manuscript and have carefully addressed each and every single one, conducting additional literature reviews to ensure thorough updates.
>
>
>
> ### RC1: Limitations of Current Methodologies
> **Reviewer request:** *"Could the author discuss limitations of current methodologies across each dimension of efficiency-focused research? For instance, the author could add another row or column in tables (e.g., table 1) to illustrate the limitation and future direction."*
>
> **Response:** We agree that surfacing limitations alongside method descriptions provides a much more actionable guide for readers. We revisited the core literature to extract these constraints and have added a **"Limitation"** column to all four summary tables (Tables 1–4).
>
> By way of example, we now explicitly document:
> * **Table 1 (Dataset):** How scaling laws become unruly in data-constrained environments.
> * **Table 2 (Architecture):** The performance gap in extreme compression ratios (e.g., *2Bits of Protein*, *CHEAP*).
> * **Table 3 (Training):** The risks of inherited teacher biases and lost conformational diversity in cross-distillation.
> * **Table 4 (Inference):** The severe accuracy degradation caused by activation quantization in models like *PTQ4Protein*.
>
> ### RC2: Model Citations and Evolution of pLMs
> **Reviewer request:** *"I noticed that many work listed in Figure 2 were not cited properly in the manuscript. It helps the clarity to add another section... to discuss the evolution and importance of protein language models and meanwhile appropriately citing milestone papers referred in Figure 2. In addition, crucial applications... could be highlighted"*
>
> **Response:** We appreciate this observation. To provide better historical context and ensure every model in Figure 2 is properly integrated, we have added a new section, **Section 1.1 (Evolution and Applications of Protein Language Models)**:
> * **Section 1.1.1** now provides a chronological narrative of the models in Figure 2 (2019–2025). It systematically traces the field from the masked-language-model era (e.g., ESM-1b, ProtTrans) through scaling and diversification, up to recent multimodal generation models like ESM3, properly citing each milestone.
> * **Section 1.1.2** highlights key applications of pLMs. We trace how downstream tasks, such as structure prediction, variant effect scoring, and functional protein generation, have evolved in complexity alongside the architectures.
>
> ### RC3: The Sequence–Structure Imbalance
> **Reviewer request:** *"This seq-struct imbalance is a significant topic of interest and naturally falls under the category of efficient dataset methods. It would be helpful to discuss how existing research adopt strategies to tackle this mismatch when training multimodal pLMs."*
>
> **Response:** We fully agree this is a central challenge for efficient pLM development. First, we updated our PDB statistic in the Introduction to reflect the current state of the field (~250,000 structures).
>
> Second, we expanded **Section 2.1 (Background and Challenges)** with a dedicated discussion on the sequence–structure imbalance. We detail three concrete strategies the field uses to train multimodal pLMs despite this gap: (1) Computational structure prediction such as treating high-confidence predicted structures (e.g., AlphaFoldDB) as training targets to close the data gap (e.g., ProstT5). (2) Cross-modal adapters such as injecting structural knowledge into frozen pLM backbones via predicted structures without requiring additional experimental data (e.g., SES-Adapter). (3) Evolutionary proxy signals such as using MSA-derived evolutionary signals as a substitute for structural information under extreme data scarcity (e.g., FSFP).
>
>
> We believe these targeted additions make the survey significantly stronger. We hope these revisions fully address your concerns and thank you again for helping us improve the paper. Please let us know if you would like us to make any other changes.

---

> > ### Comment · Reviewer_ScJL · 2026-04-02
> >
> > I thank the authors for addressing my previous questions. The addition of Section 1.1 significantly improves clarity and better positions the paper’s contributions. The related work section (1.1) could be further strengthened by including recent work on the design space of multimodal protein language models, such as [Hsieh et al.](https://arxiv.org/abs/2504.11454) (ICML 2025), which would complement the discussion of DPLM-2 and improve coverage of the broader design space. Overall, I believe this paper serves as a valuable introductory reference for researchers interested in protein language models.

---

> > > ### Author Response · Authors · 2026-04-02
> > > **Thank you**
> > >
> > > Dear Reviewer `ScJL`, we thank you for your kind words and this helpful suggestion and have now added Hsieh et al. (ICML 2025) to Section 1.1 to complement our discussion of DPLM-2 with a broader perspective on the design space of multimodal protein language models. Thank you for helping us make our paper better, please let us know if there are any additional changes you would recommend. We are committed to making this paper a detailed insightful resource for the research community.

---

### Decision · Action_Editor_zPZw · 2026-05-31

**Recommendation:** Accept as is

**Audience:**

Yes

**Audience Explanation:**

Protein language models are an active research area at the intersection of machine learning and computational biology, or more broadly AI for Science.

**Claims And Evidence:**

Yes

**Claims Explanation:**

The paper provides a comprehensive overview of efficient protein language models, and the authors have adequately addressed the reviewers' concerns regarding coverage, limitations, and positioning.

---

> ### Author Response · Authors · 2026-06-29
> **Thanks from the Authors**
>
> Dear Action Editor,
>
> Thank you very much for handling our submission and for the positive decision, as well as for awarding the Survey Certification — we are truly grateful and honored.
>
> We also sincerely appreciate your careful coordination of the review process. In particular, your help in following up with the reviewers to keep the process moving forward was invaluable, and we are grateful for the time and effort you devoted to seeing it through.
>
> We have incorporated all the reviewers' suggestions and have prepared and submitted the camera-ready version accordingly.
>
> Thank you again for your support throughout. Wishing you all the best!
>
> Best regards,
> The Authors